# ParaBlock: Communication-Computation Parallel Block Coordinate Federated Learning for Large Language Models

**Yujia Wang**                                                                        *yjw5427@psu.edu*
*College of Information Sciences and Technology*
*Pennsylvania State University*

**Yuanpu Cao**                                                                        *ymc5533@psu.edu*
*College of Information Sciences and Technology*
*Pennsylvania State University*

**Jinghui Chen**                                                                      *jzc5917@psu.edu*
*College of Information Sciences and Technology*
*Pennsylvania State University*

**Reviewed on OpenReview:** *https://openreview.net/forum?id=Hnf7eCdBeV*

## Abstract

Federated learning (FL) has been extensively studied as a privacy-preserving training paradigm. Recently, federated block coordinate descent scheme has become a popular option in training large-scale models, as it allows clients to train only a subset of the model locally instead of the entire model. However, in the era of large language models (LLMs), even a single block can contain a significant number of parameters, posing substantial communication latency, particularly for resource-constrained clients. To address this challenge in federated training/fine-tuning LLMs, we propose ParaBlock, a novel approach that establishes two parallel threads for communication and computation to enhance communication efficiency. We theoretically prove that the proposed ParaBlock achieves the same convergence rate as the standard federated block coordinate descent methods. Empirical evaluations on fine-tuning LLMs on general instruction following and mathematical reasoning confirm that ParaBlock not only maintains strong performance but also significantly improves communication efficiency.

## 1 Introduction

Federated learning (FL) (McMahan et al., 2017) is known as a promising privacy-preserving paradigm, as it keeps data on local clients without exposing sensitive information. A typical FL framework consists of a central server and multiple local clients operating in a single-threaded design. In this setup, clients repeatedly train local models, transmit the updates to the server for aggregation, and wait for the updated model from the server to continue training in the next round. While standard FedAvg type frameworks (McMahan et al., 2017; Karimireddy et al., 2020; Li et al., 2020; Wang & Ji, 2023; Wang et al., 2024b) have demonstrated success across various applications (Koutsoubis et al., 2024; Beutel et al., 2020), federated block coordinate descent schemes (Liu et al., 2019; Wu et al., 2021; Liu et al., 2024; Wang et al., 2024a) have recently gained increasing attention. Such schemes integrate local block coordinate descent (BCD) update strategies into FL, allowing clients to train only a subset of the model (commonly a block) locally instead of the entire model. Although federated BCD still operates in a single-thread manner, where clients need to wait for the model transmission to complete before the next computation step, such communication time is often negligible when models are smaller and the block size is limited.

However, with the recent trend of billion-scale large language models (LLMs), such as GPT-3 (Brown et al., 2020), Llama 3 (AI@Meta, 2024) and Gemma 2 (Team et al., 2024)), even a single layer can encompass

tens to hundreds of millions of parameters. This massive scale significantly increases the communication time between the server and clients. While such communication time was considered negligible in traditional federated block coordinate methods, it has now become a notable factor when fine-tuning LLMs on edge clients. This raises a critical bottleneck for fine-tuning LLMs with federated block coordinate descent methods: for clients with limited network bandwidth or those requiring long-distance transmission, the communication latency significantly impacts the overall efficiency of FL deployment. This inefficiency of such a single-thread approach hinders the scalability of federated block coordinate descent methods for LLMs, underscoring the need for solutions to reduce communication latency for clients.

To address the aforementioned challenge, we propose ParaBlock, a federated communication-computation **Para**llel **Block** coordinate descent method that provides a simple yet effective way to improve the communication efficiency of BCD updates for fine-tuning large-scale models on resource-constrained clients. The key contributions of this paper are summarized as follows:

- We propose ParaBlock, a novel method designed to address communication latency during the fine-tuning of LLMs using federated block coordinate descent. ParaBlock replaces the traditional single-threaded design with two parallel threads for communication and computation, effectively reducing communication delays and improving efficiency.

- We rigorously show that the proposed ParaBlock algorithm achieves a convergence rate of $\mathcal{O}(1/\sqrt{T})$ for non-convex objectives. This rate is consistent with that of standard federated block coordinate descent methods, ensuring that the improvements of ParaBlock in communication efficiency do not compromise its convergence performance.

- We conduct extensive experiments across various models and tasks to empirically validate the effectiveness of ParaBlock. Specifically, we evaluate the downstream performance of ParaBlock on general instruction following and mathematical reasoning tasks. Our results demonstrate that ParaBlock significantly reduces wall-clock runtime while maintaining performance on par with standard federated block coordinate descent baselines, highlighting its ability to enhance efficiency without compromising utility.

## 2 Related Works

**BCD for FL** Block Coordinate Descent (BCD) has been extensively studied for optimization problems (Tseng, 2001; Nesterov, 2012; Beck & Tetruashvili, 2013), particularly due to its efficiency in solving large-scale optimization tasks. Recent advancements in BCD variants, such as those proposed by Luo et al. (2024); Pan et al. (2024), further highlight its adaptability and effectiveness, especially in the context of large language models. Federated BCD schemes (Liu et al., 2019; 2024; Wu et al., 2021) enhance communication efficiency by assigning each client the responsibility for a partition of the model in a collaborative training framework. FedCyBGD (Wang et al., 2024a) focuses on fine-tuning large language models by cyclically electing one client at a time to train its assigned block of the model, enabling efficient fine-tuning with limited computational resources.

**Communication-efficiency for FL** Various methods have been proposed to improve communication efficiency in federated learning, focusing on techniques such as quantization and update compression (Reisizadeh et al., 2020; Haddadpour et al., 2019; Jin et al., 2020; Jhunjhunwala et al., 2021; Wang et al., 2022; Li et al., 2024), model pruning (Li et al., 2021; Jiang et al., 2022; Isik et al., 2022), and model distillation (Wu et al., 2022). Several existing works have been shown to enhance communication efficiency specifically in the context of federated fine-tuning of LLMs. For example, many applications of PEFT in FL help reduce communication costs (Liu et al., 2019; 2024). Additionally, FedMKT (Fan et al., 2024) leverages a mutual knowledge transfer framework, where clients and the server exchange knowledge datasets instead of model updates. Similarly, CG-FedLLM (Wu et al., 2024) introduces an encoder on clients to compress gradient features, with a corresponding decoder on the server to reconstruct the transmitted features.

**FL for LLMs** In the era of LLMs, FL has evolved to develop more complex fine-tuning and deployment of large language models (Zhang et al., 2024; Ye et al., 2024). Several existing approaches explore the use

of Parameter Efficient Fine-Tuning (PEFT) techniques for federated fine-tuning of LLMs. These include leveraging low-rank adapters (LoRA) in methods such as FedIT (Zhang et al., 2024), the OpenFedLLM framework (Ye et al., 2024), HetLoRA (Cho et al., 2024), SLoRA (Babakniya et al., 2023), FLoRA (Wang et al., 2024d), FlexLoRA (Bai et al., 2024) as well as employing adapters and bias-tuning in FedPEFT (Sun et al., 2022). In addition to the widely adopted PEFT methods in FL, recent work, such as FedCyBGD (Wang et al., 2024a), also highlights the potential of BCD in improving federated fine-tuning performance while managing resource limitations.

## 3 Preliminaries and Motivations

We start with a standard federated learning objective with $N$ total workers:

$$\min_{\boldsymbol{\theta} \in \mathbb{R}^d} f(\boldsymbol{\theta}) := \frac{1}{N} \sum_{i=1}^{N} f_i(\boldsymbol{\theta}) = \frac{1}{N} \sum_{i=1}^{N} \mathbb{E}_{\xi_i \sim \mathcal{D}_i}[f_i(\boldsymbol{\theta}; \xi_i)], \tag{1}$$

where $\boldsymbol{\theta} \in \mathbb{R}^d$ indicates the model parameter with $d$ dimensions, $f_i(\boldsymbol{\theta})$ represents the loss function associated with client $i$, and $\mathcal{D}_i$ denotes the local data distribution for client $i$. FedAvg (McMahan et al., 2017) is widely used for solving (1). In $t$-th global round, each participating client $i$ performs local training using standard SGD optimizers. The server periodically collects and aggregates the local models or updates to obtain a new global model.

**BCD and Block-wise update for FL** BCD usually has the same optimization objective as (stochastic) gradient descent. It iteratively optimizes over a small subset of parameters, while keeping the remaining parameters fixed. This approach makes BCD practical for large-scale model training. Federated BCD is then applied in FL by substituting BCD for the SGD optimizer in FedAvg. Consider a block partitioning of the model parameters $\boldsymbol{\theta}$ into $B$ blocks, i.e., $\boldsymbol{\theta} = [[\boldsymbol{\theta}]_1, [\boldsymbol{\theta}]_2, \ldots, [\boldsymbol{\theta}]_B]$. In federated BCD schemes, the goal is to approximately optimize the objective in (1) by updating only one block during local training, while keeping the remaining blocks unchanged across clients. For example, if block $b_t$ is assigned for training at global round $t$, the block-wise optimization for $[\boldsymbol{\theta}]_{b_t}$ is defined as:

$$\operatorname*{argmin}_{[\boldsymbol{\theta}]_{b_t}} f(\boldsymbol{\theta}) := \frac{1}{N} \sum_{i=1}^{N} f_i([\boldsymbol{\theta}]_1, \ldots, [\boldsymbol{\theta}]_{b_t}, \ldots, [\boldsymbol{\theta}]_B),$$

where $[\boldsymbol{\theta}]_{b_t} \in \mathbb{R}^{d_{b_t}}$ and $d_{b_t}$ denotes the dimension for block $b_t$ and there is $\sum_{b=1}^{B} d_b = d$. We summarize the key component of the local BCD update for federated BCD in Algorithm 1. Specifically, during global round $t$, client $i$ approximately optimizes local objective $f_i$ with the stochastic gradient of $f_i$:

$$\nabla f_i(\boldsymbol{\theta}; \xi) = \left[ \frac{\partial f_i}{\partial [\boldsymbol{\theta}]_1}, \ldots, \frac{\partial f_i}{\partial [\boldsymbol{\theta}]_{b_t}}, \ldots, \frac{\partial f_i}{\partial [\boldsymbol{\theta}]_B} \right]^T, \tag{2}$$

where the partial stochastic gradient for block $b_t$ is denoted as $[\boldsymbol{g}^i]_{b_t} = [\nabla f_i(\boldsymbol{\theta}; \xi)]_{b_t}$. The local model training could be summarized in Line 4 in Algorithm 1. After completing $K$ local steps of training, the client calculates the difference in local updates as shown in Line 6. The client then sends $\boldsymbol{\Delta}^i$ to the server to update the global model, and the server sends back the global model to clients for next round of local training.

**The communication bottleneck for federated BCD** While federated BCD has been applied to large-scale FL training, it still faces subsequent communication inefficiency in the era of LLMs. With the significantly larger model size and the accompanied large communication time, the current single-thread design in the federated BCD method (communication-computation-communication, as shown in Figure 1), has led to increased overall runtime due to the outstanding communication latency. This prolonged communication time forces clients to wait extensively before receiving the latest global information and proceeding to the next round of local computation. Such communication latency poses a critical bottleneck during the deployment of federated BCD, motivating us to explore new methods for improving communication efficiency and the scalability of federated BCD. Drawing inspiration from distributed training paradigms, if clients can change from the single thread design to two parallel threads that overlap the local computation with the model communication, then the communication latency can be ideally eliminated.

# 4 ParaBlock: A new federated fine-tuning method

To achieve this communication computation overlapping in federated BCD methods, we introduce ParaBlock, a **Para**llel **Block** coordinate descent method designed for fine-tuning LLMs in federated learning. ParaBlock aims to parallelize the communication thread and computation thread on local clients (as shown in Figure 1), enabling clients to perform local updates while concurrently communicating with the server. This design ensures that clients no longer need to wait for communication to finish, significantly reducing clients' idle time and accelerating the federated fine-tuning process. We summarize the proposed ParaBlock in Algorithm 2. In a nutshell, the communication

---

**Algorithm 1** LocalBlockTraining

**Input:** local learning rate $\eta_l$, global model $\boldsymbol{\theta}$ with $B$ blocks, number of local steps $K$, assigned block $b$

1: Initialize model $\boldsymbol{\theta}^i = \boldsymbol{\theta}$ on client $i$
2: **for** $k = 0$ to $K - 1$ **do**
3:    Compute local partial stochastic gradient $[\boldsymbol{g}_k^i]_b = [\nabla f_i(\boldsymbol{\theta}_k^i; \xi)]_b$
4:    Local update: $\boldsymbol{\theta}_{k+1}^i \leftarrow \boldsymbol{\theta}_k^i, \quad [\boldsymbol{\theta}_{k+1}^i]_b \leftarrow [\boldsymbol{\theta}_k^i]_b - \eta_l[\boldsymbol{g}_k^i]_b$
5: **end for**
6: Client gets $\boldsymbol{\Delta}^i = [\boldsymbol{\theta}_K^i]_b - [\boldsymbol{\theta}^i]_b$

**Output:** $\boldsymbol{\Delta}^i$

---

thread and computation thread proceed *in parallel*. Specifically, for each client $i$, the computation thread fine-tunes block $b_t$ using the LocalBlockTraining summarized in Algorithm 1. During this computation, except for the first round when $t = 0$, client $i$ proceeds with the communication thread to synchronize with the server regarding the model update variables from the previous round (*one round behind*). This involves sending $\boldsymbol{\Delta}_{t-1}^i$ to the server and receiving the aggregated $\boldsymbol{\Delta}_{t-1}$ from the server. Once both threads are completed, client $i$ immediately updates its local model for the next round of training, as illustrated in Line 9. Note that the new local model $\boldsymbol{\theta}_{t+1}^i$ inherits most of the parameters from the current model $\boldsymbol{\theta}_t^i$ but incorporates two key update steps:

1. The latest fine-tuned block $b_t$ of the local model is updated with $\boldsymbol{\Delta}_t^i$, ensuring that $[\boldsymbol{\theta}_{t+1}^i]_{b_t}$ is updated with the latest local information before starting the next computation thread.

2. The client uses the received global variable $\boldsymbol{\Delta}_{t-1}$ to correct the previous update on the local block $b_{t-1}$ by applying a correction $[\boldsymbol{\theta}_{t+1}^i]_{b_{t-1}} \leftarrow [\boldsymbol{\theta}_t^i]_{b_{t-1}} + \eta(\boldsymbol{\Delta}_{t-1} - \boldsymbol{\Delta}_{t-1}^i)$. This is because the previous update on block $b_{t-1}$ only used the local update $\boldsymbol{\Delta}_{t-1}^i$ (since the global update $\boldsymbol{\Delta}_{t-1}$ is one round behind and has not arrived yet at that time), and now we want it to be consistent with the global model by replacing the local update with the global one.

After two blocks are updated in $\boldsymbol{\theta}_{t+1}^i$, client $i$ begins the new communication and computation thread for global round $t + 1$. Therefore, unlike standard federated BCD methods and

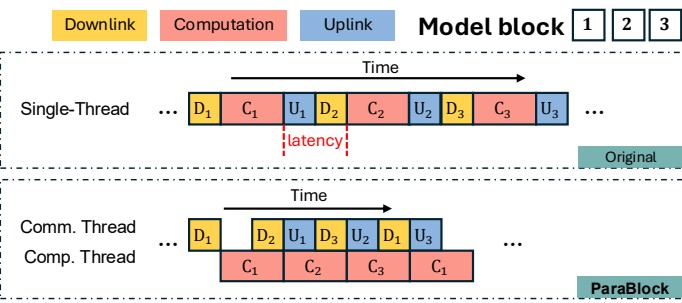

Figure 1: Comparison between the original federated BCD and the proposed ParaBlock. The original BCD's single-thread approach leads to higher runtime due to communication latency, while ParaBlock improves efficiency by overlapping communication and computation.

most traditional single-thread FL paradigms, the two-thread parallel design of ParaBlock allows clients to continue local fine-tuning while conducting communication. Although the global information is updated to the local model with one-round delay, the impact caused by this delay can be effectively mitigated by correction of ParaBlock.

Moreover, while clients update the local model, the server simultaneously updates the global model using the aggregated $\boldsymbol{\Delta}_{t-1}$ (Line 10 in Algorithm 2). Compared with standard federated BCD methods, here the global model $\boldsymbol{\theta}_t$ at round $t$ exhibits one-round staleness because the most recently trained block $b_t$ has not yet been incorporated into the global model. Due to this staleness, an extra communication and aggregation

step is required at the end of the fine-tuning (Lines 12-14) to ensure that the final computation results are aggregated and updated to the global model, then the server obtains the final model $\boldsymbol{\theta}_T$.

---

**Algorithm 2** ParaBlock

**Input:** local learning rate $\eta_l$, global learning rate $\eta$, number of block partition $B$

1: Initialize global model $\boldsymbol{\theta}_0$ and generate a block partition $b = 1, 2, \ldots, B$
2: **for** $t = 0$ to $T - 1$ **do**
3:     Each client $i$ runs a compute thread and a communication thread **in parallel**:
4:     **Compute thread:**
    $\boldsymbol{\Delta}_t^i \leftarrow \texttt{LocalBlockTraining}\left(\boldsymbol{\theta}_t^i, \eta_l, K, b_t\right)$

5:     **Communication thread:**
6:     **if** $t > 0$ **then**
7:         Client $i$ sends $\boldsymbol{\Delta}_{t-1}^i$ to the server and waits for the aggregated $\boldsymbol{\Delta}_{t-1}$
8:     **end if**
9:     // **When both threads finish, client $i$ processes:**
    updating local model: $\boldsymbol{\theta}_{t+1}^i \leftarrow \boldsymbol{\theta}_t^i$, $[\boldsymbol{\theta}_{t+1}^i]_{b_t} \leftarrow [\boldsymbol{\theta}_t^i]_{b_t} + \eta\boldsymbol{\Delta}_t^i$
    **if** $t > 0$ **then**
        $[\boldsymbol{\theta}_{t+1}^i]_{b_{t-1}} \leftarrow [\boldsymbol{\theta}_t^i]_{b_{t-1}} + \eta\boldsymbol{\Delta}_{t-1} - \eta\boldsymbol{\Delta}_{t-1}^i$
10:     **Server** maintains the global model
    $\boldsymbol{\theta}_t \leftarrow \boldsymbol{\theta}_{t-1}$, $\quad [\boldsymbol{\theta}_t]_{b_{t-1}} = [\boldsymbol{\theta}_{t-1}]_{b_{t-1}} + \eta\boldsymbol{\Delta}_{t-1}$
11: **end for**
12: Client $i$ send $\boldsymbol{\Delta}_{T-1}^i$ to the server and wait for the aggregated $\boldsymbol{\Delta}_{T-1}$
13: Server updates the global model
    $\boldsymbol{\theta}_T \leftarrow \boldsymbol{\theta}_{T-1}$, $\quad [\boldsymbol{\theta}_T]_{b_{T-1}} = [\boldsymbol{\theta}_{T-1}]_{b_{T-1}} + \eta\boldsymbol{\Delta}_{T-1}$
14: Output $\boldsymbol{\theta}_T$

---

In the following, we will mathematically demonstrate how the correction in Line 9 addresses the inconsistency in local models and we provide an analysis of the relationship between the global and local models.

**Recursive derivation of local and global models** We begin with round $t - 1$, where client $i$ continues fine-tuning using the local model $\boldsymbol{\theta}_t^i$, where $[\boldsymbol{\theta}_t^i]_{b_{t-1}} \leftarrow [\boldsymbol{\theta}_{t-1}^i]_{b_{t-1}} + \eta\boldsymbol{\Delta}_{t-1}^i$. At this stage, the model parameters of block $b_{t-1}$ differ across all clients, as they have not yet finished the synchronizing the latest updates with the server. Then by global round $t$, as the synchronization between the server and clients regarding block $b_{t-1}$ is completed, clients can correct block $b_{t-1}$ in $\boldsymbol{\theta}_{t+1}^i$ using $\boldsymbol{\Delta}_{t-1} - \boldsymbol{\Delta}_{t-1}^i$:

$$[\boldsymbol{\theta}_{t+1}^i]_{b_{t-1}} = [\boldsymbol{\theta}_t^i]_{b_{t-1}} + \eta(\boldsymbol{\Delta}_{t-1} - \boldsymbol{\Delta}_{t-1}^i) = [\boldsymbol{\theta}_{t-1}^i]_{b_{t-1}} + \eta\boldsymbol{\Delta}_{t-1}$$

$$= \ldots = [\boldsymbol{\theta}_0^i]_{b_{t-1}} + \eta\sum_{s=0}^{t-1}\mathbb{I}_s(b_{t-1})\boldsymbol{\Delta}_s = [\boldsymbol{\theta}_0]_{b_{t-1}} + \eta\sum_{s=0}^{t-1}\mathbb{I}_s(b_{t-1})\boldsymbol{\Delta}_s, \tag{3}$$

where $\mathbb{I}_s(b_{t-1})$ is an indicator function that equals to 1 if block $b_{t-1}$ is assigned for round $s$, and 0 otherwise. This recursive derivation shows that the local model for block $b_{t-1}$ can be expressed as the initial global model $\boldsymbol{\theta}_0$ combined with all relevant global updates. Thus, the one-round delayed $\boldsymbol{\Delta}_{t-1}$ corrects the inconsistent parameters in block $b_{t-1}$ of local models. This ensures that the inconsistent parameters in block $b_{t-1}$ are effectively corrected after one global round, making sure that most of the parameters across local models align in a consistent direction for subsequent local training/fine-tuning.

Similarly, for the global model $\boldsymbol{\theta}_t$, there is

$$[\boldsymbol{\theta}_t]_{b_{t-1}} = [\boldsymbol{\theta}_{t-1}]_{b_{t-1}} + \eta\boldsymbol{\Delta}_{t-1} = \ldots = [\boldsymbol{\theta}_0]_{b_{t-1}} + \eta\sum_{s=0}^{t-1}\mathbb{I}_s(b_{t-1})\boldsymbol{\Delta}_s. \tag{4}$$

This indicates that the global model's block $b_{t-1}$ can also be recursively expressed as the initial global model with relevant updates. Based upon the result in Eq. (3) and Eq. (4), the block $b_{t-1}$ of local model $\boldsymbol{\theta}_{t+1}^i$ exactly matches the block $b_{t-1}$ in global model $\boldsymbol{\theta}_t$. This demonstrates the effectiveness of the model correction mechanism in ParaBlock.

**Discussion about privacy protection** While FL ensures data ownership protection (raw data stays local), it does not automatically prevent the model from learning and outputting sensitive patterns. This privacy concern is a known challenge in generative language models. ParaBlock primarily addresses the efficiency and communication bottlenecks of training LLMs in this distributed setting. Thus, ParaBlock would be orthogonal to advanced privacy protection techniques like data anonymization or differential privacy to minimize the risk of data privacy leakage.

## 5 Theoretical Analysis

In this section, we delve into the convergence guarantee of the proposed ParaBlock algorithm. We begin by outlining the key assumptions necessary for the analysis. Following this, we present the convergence rate and provide a detailed discussion of the results.

**Assumption 5.1** (Smoothness). The local objective function $f_i(\boldsymbol{\theta})$ is $L$-smooth, i.e., $\forall \boldsymbol{\theta}_1, \boldsymbol{\theta}_2 \in \mathbb{R}^d$,

$$\|\nabla f_i(\boldsymbol{\theta}_1) - \nabla f_i(\boldsymbol{\theta}_2)\| \leq L\|\boldsymbol{\theta}_1 - \boldsymbol{\theta}_2\|.$$

**Assumption 5.2** (Bounded Variance). The stochastic gradient computed on local client is unbiased and has a bounded local variance, i.e., for all $\boldsymbol{\theta}$ and $i \in [N]$, we have $\mathbb{E}\big[\|\nabla f_i(\boldsymbol{\theta};\xi) - \nabla f_i(\boldsymbol{\theta})\|^2\big] \leq \sigma^2$, and the loss functions has a global variance bound, $\frac{1}{N}\sum_{i=1}^N \|\nabla f_i(\boldsymbol{\theta}) - \nabla f(\boldsymbol{\theta})\|^2 \leq \sigma_g^2$.

Assumption 5.1 and 5.2 are common assumptions in analyzing federated non-convex optimization methods (Li et al., 2019; Yang et al., 2021; Reddi et al., 2021; Wang et al., 2022; Wang & Ji, 2023; Wang et al., 2024c). The global variance upper bound of $\sigma_g^2$ in Assumption 5.2 measures the data heterogeneity across clients, where $\sigma_g^2 = 0$ indicates i.i.d. data distribution across clients.

In the following, we present the theoretical convergence analysis of ParaBlock. For clarity and fair comparison with existing analyses of FedBCD methods, we conduct the analysis under the local SGD optimizer. Extensions to local adaptive optimizers, along with *additional discussions on the connection between general and block-wise properties* are provided in Appendix B.

**Theorem 5.3.** *Under Assumptions 5.1–5.2, let $T$ represent the total number of global rounds, $K$ be the number of local SGD training steps and $N$ be the number of the clients. If the learning rate $\eta$ and $\eta_l$ satisfy $\eta_l \leq \frac{1}{22KL}$ and $\eta\eta_l \leq \frac{1}{4KL}$ , then the global iterates $\{\boldsymbol{\theta}_t\}_{t=0}^{T-1}$ of Algorithm 2 satisfy*

$$\frac{1}{T}\sum_{t=0}^{T-1} \mathbb{E}[\|\nabla_{b_t} f(\boldsymbol{\theta}_t)\|^2] \leq \frac{8\mathcal{F}}{\eta\eta_l TK} + 40\eta_l^2 L^2 K(\sigma^2 + 6K\sigma_g^2)$$

$$+ \left(8\eta^2\eta_l L^2 K + \frac{\eta L}{2}\right)\frac{\eta_l}{N}\sigma^2 + 64\eta_l^2\eta^2 L^2 K[\sigma^2 + 10\eta_l^2 L^2 K^2(\sigma^2 + 6K\sigma_g^2)], \qquad (5)$$

*where $\mathcal{F} = f(\boldsymbol{\theta}_0) - f_*$ and $f_* = \min_{\boldsymbol{\theta}} f(\boldsymbol{\theta}) > -\infty$.*

**Corollary 5.4.** *If we choose the global learning rate $\eta = \Theta(\sqrt{KN})$ and $\eta_l = \Theta\big(\frac{1}{\sqrt{TK}}\big)$ in Theorem 5.3, then for sufficiently large $T$, the global iterates $\{\boldsymbol{\theta}_t\}_{t=0}^{T-1}$ of Algorithm 2 satisfy*

$$\frac{1}{T}\sum_{t=0}^{T-1} \mathbb{E}[\|\nabla_{b_t} f(\boldsymbol{\theta}_t)\|^2] \leq \mathcal{O}\left(\frac{\mathcal{F} + \sigma^2}{\sqrt{TKN}} + \frac{N\sigma^2 + \sigma_g^2}{T}\right). \qquad (6)$$

*Remark* 5.5. Compared with the existing federated BCD methods such as FedBCD (Liu et al., 2019; Wu et al., 2021) and FedBCGD (Liu et al., 2024), our proposed method obtains the same $\mathcal{O}(1/\sqrt{T})$ convergence rate under general non-convex settings. This demonstrates that the one round staleness introduced by the communication-computation parallel thread in ParaBlock does not compromise its overall convergence guarantee.

## 6 Experiments

In general, the proposed method leverages local block updates combined with a communication-computation parallel scheme during fine-tuning. This section presents a series of experiments to evaluate both the

performance and efficiency of our approach. First, we assess the performance of ParaBlock and compare it against several existing federated fine-tuning algorithms. Next, we analyze the time efficiency of ParaBlock under varying network conditions and computational demands. Finally, we perform ablation studies to gain deeper insights into the key factors influencing the effectiveness of ParaBlock.

### 6.1 Experimental settings

We summarize some crucial implementation details in the following, and we leave some additional results and experiment details to Appendix A.

**Datasets, models, and evaluations** We utilize the Alpaca-GPT4 dataset (Peng et al., 2023) for general instruction following tasks, and we use 50000 data samples sampled from the MathInstruct dataset (Yue et al., 2023) for mathematical reasoning task. We fine-tune two models, Llama 3-8B (Dubey et al., 2024; AI@Meta, 2024) and a lightweight Llama 3.2-3B (AI@Meta, 2024) designed for on-device use. To assess the performance when fine-tuning on instruction following task, we utilize MT-Bench (Zheng et al., 2023) with GPT-4o as a judge model. For evaluating the performance on mathematical reasoning, we employ the widely used OpenLLM Leaderboard (Beeching et al., 2023) as the evaluation benchmark and we report the evaluation score on GSM8K (Cobbe et al., 2021) to show the math problem solving capability.

Table 1: Fine-tune Llama 3-8B and Llama 3.2-3B (AI@Meta, 2024) on Alpaca-GPT4 dataset (Peng et al., 2023) and MathInstruct dataset (Yue et al., 2023). In our results, we highlight the best score in **bold** and the second-best score with an underline.

| Method | Alpaca-GPT4 | | | | Math Instruct | | | |
|---|---|---|---|---|---|---|---|---|
| | Llama 3-8B | | Llama 3.2-3B | | Llama 3-8B | | Llama 3.2-3B | |
| | MT-Bench↑ | RT(m)↓ | MT-Bench↑ | RT(m)↓ | GSM8K↑ | RT(m)↓ | GSM8K↑ | RT(m)↓ |
| Base | 4.72 | - | 4.18 | - | 51.55 | - | 27.98 | - |
| Fed full FT | **5.33** | **16.0** | 4.36 | 12.5 | 55.80 | **15.5** | **32.22** | 12.3 |
| FedIT | 5.11 | 34.5 | 4.31 | 23.8 | 54.60 | 23.1 | 29.87 | 15.4 |
| FFA-LoRA | 5.08 | 30.2 | 4.25 | 23.3 | 54.59 | 21.2 | 30.55 | 14.9 |
| FLoRA | 4.95 | 65.4 | 4.23 | 34.0 | 52.54 | 64.9 | 28.51 | 29.3 |
| FedCyBGD | 4.87 | 59.9 | 4.29 | 57.8 | 51.40 | 63.2 | 27.60 | 53.3 |
| FedBCD | 5.14 | 30.2 | 4.33 | 19.1 | 54.74 | 24.9 | 31.84 | 17.3 |
| ParaBlock | 5.14 | 21.1 | **4.40** | **11.9** | **55.88** | 15.8 | 31.77 | **10.1** |

**Baselines** We compare the ParaBlock with several federated fine-tuning baselines including 1) Federated full model fine-tuning (Fed full FT), 2) FedIT (Zhang et al., 2024), which is one of the most commonly used federated fine-tuning methods that integrates LoRA (Hu et al., 2021) into standard FedAvg (McMahan et al., 2017) method, 3) FFA-LoRA (Sun et al., 2024), which fixes the LoRA matrix **B** and fine-tunes the LoRA matrix **A** to reduce server aggregation bias, 4) FLoRA (Wang et al., 2024d), a recent method for federated fine-tuning with LoRA, 5) FedCyBGD (Wang et al., 2024a), which employs the cyclic update for block coordinate federated fine-tuning, and 6) FedBCD, a standard federated BCD scheme that directly integrates local BCD updates into the original FedAvg aggregation schemes. We provide the GPU consumption of all baselines in Table 9 in Appendix A.

**Implementation details** We implement federated fine-tuning of LLMs by setting up an FL framework with 10 clients, each assigned a local dataset. Notably, we perform heterogeneous data partitioning for both datasets. For the Alpaca-GPT4 dataset, we adopt a text clustering method similar to the one used in Lin et al. (2021) to get a cluster label. For the MathInstruct dataset, we use the "source" from the original dataset as a label and follow traditional data partitioning using a Dirichlet distribution as described in Wang et al. (2020b;a). Specifically, we adopt Dirichlet(0.1) for the Alpaca-GPT4 dataset and Dirichlet(0.6) for the MathInstruct dataset. Regarding LoRA-related methods, the LoRA rank is set to 32 for FedIT, FFA-LoRA and FLoRA. For FedCyBGD, the model is partitioned into 10 blocks, with each block assigned to a corresponding client, and clients perform fine-tuning sequentially. For FedBCD and the proposed ParaBlock, the model is partitioned into 16 blocks for Llama 3-8B and 14 blocks for Llama 3.2-3B. The block partition is based on the default layer of the language model. During each global round, the server randomly selects one

block for fine-tuning. We conduct 32 global rounds for fine-tuning Llama 3-8B and 28 rounds for fine-tuning Llama 3.2-3B for all BCD-based and LoRA-based baselines, and we conduct 3 global rounds for Fed full FT. We adopt AdamW as the local optimizer, i.e., conducting local block training via AdamW, as it is the default optimizer for most of LLMs training and fine-tuning. The default effective batch size in our experiment is set to 4. The random seeds for all libraries is 42. All experiments are conducted on NVIDIA A100 GPUs. To fully support reproducibility, our code will be released later.

## 6.2 Main results

**Results on general instruction following task**   We begin by evaluating the performance of the proposed ParaBlock on the general instruction-following dataset, Alpaca GPT-4 (Peng et al., 2023), for language model fine-tuning. We report the MT-bench score for evaluation. As shown in the results for Alpaca-GPT4 in Table 1, ParaBlock achieves a lower score than Fed full FT but consistently outperforms most LoRA-based and BCD-based PEFT methods across two models. For the Llama 3-8B model, ParaBlock achieves an MT-bench score of 5.14, which is the same score as the FedBCD, and ParaBlock shows substantial improvement over FedCyBGD, another federated block-coordinate method. Additionally, ParaBlock significantly outperforms LoRA-based FL methods such as FedIT, FFA-LoRA and FLoRA. A key highlight of ParaBlock is its *superior runtime efficiency*, as it requires outstanding less runtime compared to other baselines, particularly LoRA-based FL methods, while achieving notable performance improvements. Similarly, when fine-tuning the lightweight Llama 3.2-3B model, ParaBlock achieves the best performance among all baselines. Its ability to surpass competing methods in both performance and time efficiency highlights the effectiveness of ParaBlock in fine-tuning LLMs for general instruction-following tasks.

**Results on mathematical reasoning task**   We evaluate ParaBlock on mathematical reasoning tasks using the MathInstruct (Yue et al., 2023) dataset, with GSM8K (Cobbe et al., 2021) as the benchmark for evaluation. As shown in the right main columns of Table 1, ParaBlock outperforms all baseline methods and achieves the second-best runtime on the 8B model, and outperforms most baselines with the less runtime on the 3B model. For 8B model, ParaBlock demonstrates improvements over FedBCD, while for 3B model, it is slightly less accurate than FedBCD. This indicates that while the communication thread of ParaBlock exists one round behind, it does not compromise overall performance. ParaBlock also consistently exhibits strong performance compared to other baselines, with a notable 11.7% improvement over FedCyBGD when fine-tuning the lightweight 3B model. Furthermore, ParaBlock keeps its advantage of runtime saving among all baselines, highlighting its ability to reduce communication latency for federated BCD schemes and achieve communication efficiency in fine-tuning LLMs.

**Time efficiency**   We conduct a detailed comparison of the runtime and emphasize the time efficiency benefits achieved by leveraging communication-computation parallelism, as illustrated in Figure 2. To evaluate the necessity and benefits of this scheme, we measure wall-clock time across different network bandwidth conditions (50M/s, 100M/s, and 150M/s) and varying effective batch sizes (2, 4, and 8) on each client. [1] Among the LoRA-based federated fine-tuning methods, FFA-LoRA requires slightly less runtime than FedIT, while FLoRA incurs significantly more runtime than FedIT. Due to space limitations, we include only the detailed runtime comparison results for FedIT here, with a comprehensive discussion of all methods provided in Appendix A.

Figure 2(a) illustrates the runtime when fine-tuning Llama 3-8B model using Alpaca GPT-4 dataset under various conditions. In scenarios with low network bandwidth and minimal client-side computation, such as a bandwidth of 50M/s and an effective batch size of 2, the communication time cannot be fully overlapped by client computation. This results a significantly extra communication overhead for ParaBlock, represented by the dark yellow portion of the histogram in Figure 2(a). As the effective batch size grows, requiring more client-side computation, a greater portion of the computation time can be overlapped with communication. We notice that for faster communication networks (100M/s and 150M/s), the computation time for ParaBlock does not introduce extra communication during fine-tuning. The negligible dark yellow portion in Figure 2(a) represents the final communication process, as described in Line 12 of Algorithm 2.

---

[1]Due to deployment constraints, we can only simulate the communication bandwidth with three network bandwidth conditions, which were frequent settings in FL and decentralized learning.

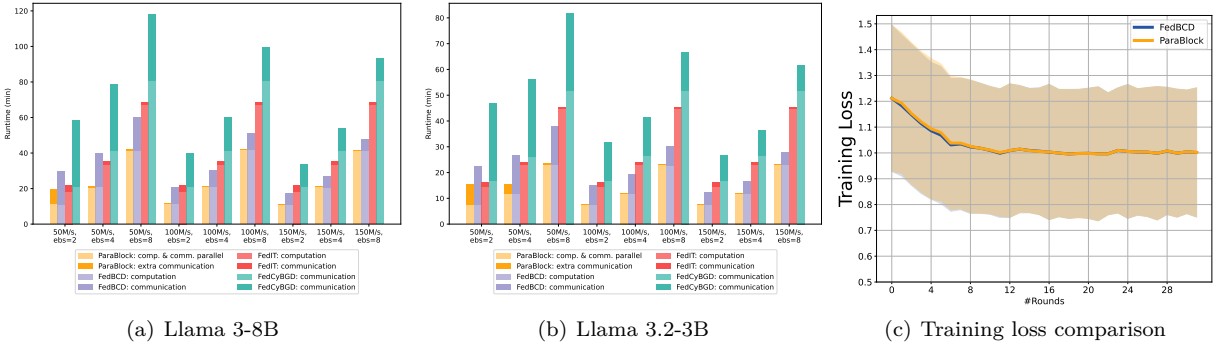

(a) Llama 3-8B                    (b) Llama 3.2-3B                    (c) Training loss comparison

Figure 2: Time efficiency: wall-clock runtime for various network communication bandwidths and effective batch sizes.

Figure 2(a) also demonstrates that ParaBlock significantly reduces runtime compared to other baselines. Notably, we emphasize the time savings achieved over block-coordinate baselines, including vanilla FedBCD and FedCyBGD. The overall runtime reduction can exceed 30% for network bandwidths of 50M/s and 100M/s with an effective batch size (ebs) of 2, with efficiency gains over FedCyBGD being particularly pronounced. It is important to note that the higher runtime costs for FedCyBGD can be attributed to several factors. First, FedCyBGD employs a cyclic update approach that selects only one client at a time, inherently prolonging the fine-tuning process. Additionally, FedCyBGD introduces extra communication overhead, as all clients must periodically synchronize to receive model updates. Thus FedCyBGD's design results in increased latency and contributes to the overall longer runtime observed in our evaluations.

The runtime savings over the LoRA-based FedIT are also notable. Figure 2(a) indicates that ParaBlock achieves comparable, and slightly better, time efficiency than FedIT under conditions of low computational cost and high communication overhead (e.g., 50M/s network and ebs=2). This is primarily due to the additional communication time associated with ParaBlock. Moreover, under most other settings, ParaBlock consistently reduces runtime by approximately 40% compared to FedIT. This improvement is attributed to the communication-computation parallelism of ParaBlock and the computational efficiency of the block coordinate method.

Figure 2(b) illustrates the fine-tuning runtime of the Llama 3.2-3B model under conditions similar to those previously described. In addition to highlighting the superior time efficiency of ParaBlock, we observe that parameter communication constitutes a larger proportion of the total runtime during the fine-tuning of the Llama 3.2-3B model. This is reflected in the increased size of the dark yellow portion in Figure 2(b), which represents a greater share of the overall runtime. These underscore the critical need to reduce communication latency for clients. By mitigating the impact of increased communication overhead, ParaBlock achieves even greater time savings. These results further demonstrate the effectiveness of ParaBlock in enhancing fine-tuning efficiency by minimizing the influence of communication delays.

We verify the convergence of the proposed ParaBlock through the training loss as shown in Figure 2(c). From an optimization perspective, ParaBlock and FedBCD exhibit very similar convergence behavior, with ParaBlock showing only a marginally slower convergence than FedBCD in the initial stages, while this gap diminishes significantly in later rounds. Note that the primary distinction between FedBCD and ParaBlock lies in the global model of ParaBlock, which introduces one-step staleness. Nevertheless, the results from Figure 2(c) indicate that this staleness has a minimal impact on the fine-tuning convergence of ParaBlock.

## 6.3 Ablation studies

We analyze several aspects of the proposed ParaBlock, including: 1) How many blocks should ParaBlock use to balance utility and time efficiency? 2) Is there any block scheduling strategy can achieve better fine-tuning results? 3) How does the data distribution among clients impact the fine-tuning performance? We further study staleness beyond one round, extend the analysis beyond LLaMA architectures, and evaluate under cross-silo partial participation. Additional results are provided in Appendix A.

**Ablation for block partition** In our experiments, as previously mentioned, we partition blocks based on the default layers in language models. To investigate the impact of block partitioning, we explore different configurations by varying the number of layers within each block in the proposed ParaBlock approach. As shown in Table 2, we evaluate configurations where each block contains partial, 1, 2, or 4 layers during fine-tuning of the Llama 3-8B model on both general and math tasks. The results in Table 2 indicate that the runtime gets longer as there are more layers assigned to one block. Note that the increase in runtime encompasses the growth of both computation and communication runtime, and we notice that assigning 2 layers per block yields slightly better performance compared to other configurations in both tasks.

Table 2: Ablation for the block assignment the number of layers.

| Models | MT-B↑ | RT(m)↓ | GSM8K↑ | RT(m)↓ |
|---|---|---|---|---|
| Partial layer | 5.12 | **20.3** | 53.22 | **15.0** |
| 1 layer | 5.08 | 20.4 | 53.90 | 15.1 |
| 2 layers | **5.14** | 21.1 | **55.88** | 15.8 |
| 4 layers | 5.13 | 22.7 | 55.04 | 19.4 |

**Ablation for block scheduling** We investigate how various block partitions would impact on the overall performance. We compare the default random scheduling which randomly selects two layers based on the default layers in Llama 3-8B, the sequentially, reverse sequentially layer scheduling and a gradient-guided scheduling based on the original gradient. As shown in Table 3, we find that Random scheduling can achieve the best result in math reasoning but performs slightly worse than gradient-based in general instruction tuning. However, the gradient-based approach incurs extra computational cost, as it requires a full-model backward pass prior to fine-tuning in order to determine the scheduling.

**Ablation for heterogeneous distribution on clients** We compare the i.i.d. and non-i.i.d. data partitions when fine-tuning on the general instruction dataset Alpaca-GPT4 (Peng et al., 2023) and math reasoning dataset Math Instruct (Yue et al., 2023), as shown in Table 4. We further conduct experiments under an extreme non-i.i.d. setting with Dirichlet(0.01) applied to both datasets. Our observations indicate that i.i.d. data sampling consistently results in higher evaluation scores for both general instruction tuning and mathematical reasoning tasks.

Table 3: Ablation for block scheduling

| Models | MT-B↑ | GSM8K↑ |
|---|---|---|
| Random | 5.14 | **55.88** |
| Seq. | 5.03 | 54.21 |
| Rev. seq. | 5.06 | 54.59 |
| Gradient based | **5.19** | 53.60 |

**Ablation for the number of staleness rounds** The one-round staleness in the original ParaBlock can be readily extended to multi-round staleness settings. For example, if we assume that all clients in the network experience lower communication bandwidth, leading to communication-computation parallelism with two rounds of staleness, we obtain the following results as shown in Table 5 . It shows a slight performance degradation when all clients incur two-round staleness; however, the performance remains comparable to the base model score.

Table 4: The results for fine-tuning LLaMA 3-8B on two datasets, considering i.i.d., non-i.i.d. and extreme non-i.i.d. partitions.

| Data distribution | MT-B↑ | GSM8K↑ |
|---|---|---|
| i.i.d. | **5.21** | **56.14** |
| Non-i.i.d. | 5.14 | 55.88 |
| Extreme non-i.i.d. | 5.13 | 54.66 |

**Experiments beyond Llama model architectures** We conduct mathematical reasoning experiments using the same MathInstruct dataset as in the main experiments with the Qwen-2.5-1.5B-Instruct model (Team, 2024), and evaluate the fine-tuned models on GSM8K. The results in Table 6 demonstrate that ParaBlock consistently outperforms other baselines in terms of accuracy, while also achieving the second-best runtime, highlighting the effectiveness of our design.

Table 6: Fine tune Qwen-2.5-1.5B-Instruct model on MathInstruct dataset. We highlight the best score in **bold** and the second-best score with an underline.

| Method | GSM8K↑ | RT(m)↓ |
|---|---|---|
| Base | 54.28 | – |
| Fed full FT | 60.58 | **9.3** |
| FedIT | 62.32 | 19.76 |
| FFA-LoRA | 61.56 | 17.12 |
| FLoRA | 57.77 | 52.42 |
| FedCyBGD | 54.51 | 42.01 |
| FedBCD | 61.79 | 18.48 |
| ParaBlock | **62.70** | 10.56 |

Table 7: Fine tune Llama 3-8B model on MathInstruct dataset with 50 clients cross-silo settings.

| Method | MT-B ↑ | RT(m)↓ | GSM8K↑ | RT(m)↓ |
|---|---|---|---|---|
| FedIT | 5.06 | 49.31 | 53.75 | 44.15 |
| FedBCD | 5.08 | 55.36 | 53.60 | 52.16 |
| ParaBlock | **5.10** | **28.45** | **53.90** | **28.44** |

**Extension to cross-silo partial participation settings** ParaBlock is fully compatible with partial participation, and we have conducted experiments demonstrating its effectiveness under such settings. Specifically, we consider a cross-silo setup where 20% of 50 clients are randomly selected in each round, and we fine-tune the Llama-3 8B model on Alpaca-GPT4 and Math Instruct dataset. We compare this cross-silo ParaBlock with FedBCD and FedIT. As shown in Table 7, ParaBlock outperforms FedIT and FedBCD while retaining its computation time-saving advantage.

Table 5: Ablation for the number of staleness rounds.

| Method | GSM8K↑ | RT(m)↓ |
|---|---|---|
| Base | 51.55 | - |
| one-round staleness | **55.88** | **15.8** |
| two-round staleness | 54.74 | 16.1 |

**Extension to heterogeneous communication bandwidth** The proposed ParaBlock is natural to extend to heterogeneous communication bandwidth. For example, assume one client was significantly slower than the others. This extremely slow client incurs a three-round staleness, while other clients maintain the original one round of staleness. The results in Table 8 illustrate this scenario. Note that in this heterogeneous bandwidth setting, slow clients with significant latency may degrade overall performance. We believe it requires a non-trivial design for better performance and conclude this as future work.

Table 8: Fine tune Llama 3-8B model on MathInstruct dataset with heterogeneous communication bandwidth.

| Method | GSM8K↑ | RT(m)↓ |
|---|---|---|
| homogeneous bandwidth (original) | 55.88 | 38.72 |
| heterogeneous bandwidth | 52.46 | 20.16 |

**Discussion on the comparison with asynchronous FL baselines** ParaBlock is significantly different from asynchronous FL, although both share the motivation of addressing communication delays and efficiency. The efficiency in asynchronous FL comes from letting clients update the server at their own pace, so it may happen that one client is conducting communication while another is computing gradients. ParaBlock, however, sets up a two-thread parallel scheme for every single client. Thus, it may not be fair to directly compare asynchronous FL approaches with ParaBlock, as they are designed for full-parameter training. Nevertheless, we compare one representative asynchronous FL baseline, FedBuff (Nguyen et al., 2022), with ParaBlock for fine-tuning the Llama 3-8B model on mathematical reasoning tasks. In a setting where 2 out of 10 clients experience extreme delays, and the FedBuff algorithm is applied, the GSM8K score drops to 52.24, which is significantly lower than ParaBlock's score of 55.88. We believe this degradation is due to the impact of slow clients in FedBuff. While asynchronous training remains a promising direction, it would require a non-trivial redesign to be effective.

# 7   Conclusion

In this paper, we propose ParaBlock, a communication-computation parallel block coordinate method for federated BCD to enhance communication efficiency. To better support this two-thread parallel method, we adjust the local model initialization and global model update based on the FL-BCD schemes. We theoretically show the convergence rate for the proposed ParaBlock under general non-convex settings, which indicates our design does not sacrifice the convergence of the standard FL-BCD. We perform extensive experiments on diverse models and tasks to empirically assess the effectiveness of ParaBlock. The results show that ParaBlock significantly improves communication efficiency while achieving performance comparable to standard FL-BCD baselines, highlighting its effectiveness in enhancing efficiency in FL deployments.

# Acknowledgments

We thank the anonymous reviewers for their helpful comments. This work is partially supported by the National Science Foundation under Grant No. 2348541. The views and conclusions contained in this paper are those of the authors and should not be interpreted as representing any funding agencies.

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

# A Additional Experiments

## A.1 Additional Results

**Additional experiments on multilingual settings** We conducted multilingual experiments by fine-tuning both the Spanish and the original English Alpaca-GPT4 models, and evaluated them on the ARC challenge. Due to time and space constraints, we only compare ParaBlock with FedBCD here to demonstrate that ParaBlock can still achieve performance comparable to FedBCD.

**GPU memory consumption** We first present the peak GPU memory consumption for all baselines in Table 9 when fine-tuning on the Alpaca-GPT4 dataset (Peng et al., 2023). The results indicate that ParaBlock and FedBCD incur the lowest GPU memory costs among the fine-tuning methods. FedCyBGD exhibits a relatively higher memory cost compared to FedBCD and ParaBlock due to differences in block assignment. Furthermore, the proposed ParaBlock consistently consumes less memory than LoRA-based methods.

Table 9: GPU peak consumption when fine-tuning the Alpaca-GPT4 dataset.

| Methods | Llama 3-8B | Llama 3.2-3B |
|---|---|---|
| FedIT | 27.0G | 14.3G |
| FFA-LoRA | 26.8G | 14.2G |
| FLoRA | 27.0G | 14.3G |
| FedCyBGD | 29.6G | 13.9G |
| FedBCD | 23.3G | 10.0G |
| ParaBlock | 23.3G | 10.0G |

**Orthogonal to existing communication efficient FL methods** Previous studies in FL have explored various communication-efficient techniques, such as model/update compression, quantization, and pruning (Reisizadeh et al., 2020; Haddadpour et al., 2019; Wang et al., 2022; Jiang et al., 2022). These approaches primarily aim to reduce the number of communication bits, thereby improving the communication efficiency in FL systems. In contrast, our proposed ParaBlock targets the latency inherent in the communication process, making ParaBlock orthogonal to existing communication-reduction methods. In Table 10, 1)we compare ParaBlock with existing communication-reduction method applied to FedBCD, and 2) we show that for cases where the computation time cannot overlap the communication time, we can further apply existing communication-reduction methods to improve the communication efficiency.

The top part of Table 10 shows that ParaBlock achieves superior performance compared to directly applying top-$k$ compression (as utilized in (Wang et al., 2022; Li et al., 2024)) with top 20% ratio to the standard FedBCD baseline, while also requiring less runtime. This is because top-$k$ compression, despite reducing communication bits, still necessitates transmitting compressed model at each global round. In contrast, ParaBlock directly reduces the communication latency, resulting in better overall performance than applying compression to the vanilla FedBCD baseline. Moreover, ParaBlock is compatible with existing communication reduction techniques, as shown in the bottom lines in Table 10. By further integrating top-$k$ compression, ParaBlock can effectively reduce the extra communication time with still achieving reasonable performance in both tasks.

**Hyper-parameter details** We conduct learning rate searches to find the best learning rate for each baseline. We perform a grid search over the learning rate $\eta_l$ from {3e-7, 1e-6, 3e-6, 1e-5, 3e-5 }, and the global learning rate $\eta = 1$ for all experiments. The extra hyper-parameters for AdamW optimizer is following the default parameter in `Trainer`, i.e., $\beta_1 = 0.9, \beta_2 = 0.999, \epsilon = 10^{-6}$. Table 11 summarizes the learning rates in our experiments.

Table 10: Comparison with Top-$k$ compression methods, where MT-B is the abbreviation for MT-Bench.

| Method | MT-B↑ | Runtime(m) ↓ | GSM8K↑ | Runtime(m) ↓ |
|---|---|---|---|---|
| | **100M/s, ebs=4** | | | |
| FedBCD | **5.14** | 30.2 | 54.74 | 24.9 |
| +Top-20% | 5.00 | 26.4 | 54.12 | 21.1 |
| ParaBlock | **5.14** | 21.1 | **55.88** | 15.8 |
| +Top-20% | 5.11 | **21.0** | 55.27 | **15.6** |
| | **50M/s, ebs=2** | | | |
| FedBCD | **5.03** | 30.0 | **54.66** | 26.8 |
| +Top-20% | 4.99 | 22.5 | 53.90 | 19.2 |
| ParaBlock | 4.99 | 19.4 | 53.53 | 19.4 |
| +Top-20% | 4.98 | **11.6** | 54.14 | **11.6** |

Table 11: Learning rates in our experiments

| | Alpaca-GPT4 | | Math Instruct | |
|---|---|---|---|---|
| | Llama 3-8B | Llama 3.2-3B | Llama 3-8B | Llama 3.2-3B |
| FFT | 3e-7 | 1e-7 | 1e-7 | 1e-6 |
| FedIT | 3e-6 | 3e-7 | 3e-6 | 3e-5 |
| FFA-LoRA | 3e-6 | 3e-7 | 3e-6 | 3e-5 |
| FLoRA | 3e-6 | 3e-7 | 3e-6 | 3e-5 |
| FedCyBGD | 1e-5 | 1e-6 | 3e-6 | 3e-5 |
| FedBCD | 1e-5 | 1e-6 | 3e-6 | 3e-5 |
| ParaBlock | 1e-5 | 1e-6 | 3e-6 | 3e-5 |

# B  Theoretical Analysis

## B.1  Additional Discussions about Assumptions

**Discussion about Assumption 5.1.**  The block-wise smoothness property could be naturally implied by the general smoothness of objective function. Given the non-negativity of the norm operation, there is $\|\nabla_b f(\boldsymbol{\theta}_1) - \nabla_b f(\boldsymbol{\theta}_2)\| \leq \|\nabla f(\boldsymbol{\theta}_1) - \nabla f(\boldsymbol{\theta}_2)\| = L\|\boldsymbol{\theta}_1 - \boldsymbol{\theta}_2\|$. We adopt the general smoothness for convenience and notational clarity. Alternatively, if we assume block-wise smoothness: for each block $b$, there is $\|\nabla_b f(\boldsymbol{\theta}_1) - \nabla_b f(\boldsymbol{\theta}_2)\| \leq L_b\|\boldsymbol{\theta}_1 - \boldsymbol{\theta}_2\|$, and with $\bar{L} = \max_b L_b$, the convergence analysis can be modified accordingly. The convergence rate will maintain $O(1/\sqrt{T})$ but depend on $\bar{L}$ instead of $L$.

**Discussion about Assumption 5.2.**  The block-wise heterogeneity naturally follows from the original bounded heterogeneity in Assumption 5.2 as well. Using the property of partial derivatives, $\nabla_b f(\boldsymbol{\theta}_t) = [\nabla f(\boldsymbol{\theta}_t)]_b$, and following the argument in Lemma C.3, we have

$$\frac{1}{N}\sum_{i=1}^{N}\sum_{b=1}^{B}\|[\nabla f(\boldsymbol{\theta})]_b - [\nabla f_i(\boldsymbol{\theta})]_b\|^2 = \frac{1}{N}\sum_{i=1}^{N}\|\nabla f(\boldsymbol{\theta}) - \nabla f_i(\boldsymbol{\theta})\|^2 \leq \sigma_g^2 \tag{7}$$

Therefore, for simplicity, we adopt a general bounded variance assumption on the full gradient. Moreover, if we instead assume bounded variances for each block, the convergence rate of $O(1/\sqrt{T})$ remains valid.

## B.2 Convergence Analysis

For the global model of two consecutive steps, there is

$$[\boldsymbol{\theta}_{t+1}]_{b_t} - [\boldsymbol{\theta}_t]_{b_t} = \eta \boldsymbol{\Delta}_t. \tag{8}$$

For $\bar{\boldsymbol{\Delta}}_t = [\mathbf{0}, \cdots, \mathbf{0}, \boldsymbol{\Delta}_t, \mathbf{0}, \cdots, \mathbf{0}]$, where $[\bar{\boldsymbol{\Delta}}_t]_{b_t} = \boldsymbol{\Delta}_t$. Given the fact that $[\nabla f(\boldsymbol{\theta}_t)]_b = \nabla_b f(\boldsymbol{\theta}_t)$, for each time step $t$,

$$
\begin{aligned}
&\mathbb{E}[f(\boldsymbol{\theta}_{t+1}) - f(\boldsymbol{\theta}_t)] \\
&= \mathbb{E}[f(\boldsymbol{\theta}_{t+1})] - f(\boldsymbol{\theta}_t) \\
&\leq \mathbb{E}[\langle \nabla f(\boldsymbol{\theta}_t), \boldsymbol{\theta}_{t+1} - \boldsymbol{\theta}_t \rangle] + \frac{L}{2}\mathbb{E}[\|\boldsymbol{\theta}_{t+1} - \boldsymbol{\theta}_t\|^2] \\
&= \mathbb{E}[\langle \nabla_{b_t} f(\boldsymbol{\theta}_t), \eta\boldsymbol{\Delta}_t \rangle] + \frac{L}{2}\mathbb{E}[\|\eta\boldsymbol{\Delta}_t\|^2] \\
&= \underbrace{\eta\mathbb{E}[\langle \nabla_{b_t} f(\boldsymbol{\theta}_t), \boldsymbol{\Delta}_t \rangle]}_{I_1} + \underbrace{\frac{\eta^2 L}{2}\mathbb{E}[\|\boldsymbol{\Delta}_t\|^2]}_{I_2}.
\end{aligned} \tag{9}
$$

where the first inequality follows Assumption 5.1, and the second equation holds by $[\nabla f(\boldsymbol{\theta}_t)]_b = \nabla_b f(\boldsymbol{\theta}_t)$ and Eq. (8). For the first term $I_1$, there is

$$
\begin{aligned}
I_1 &= \eta\mathbb{E}[\langle \nabla_{b_t} f(\boldsymbol{\theta}_t), \boldsymbol{\Delta}_t \rangle] \\
&= \eta\mathbb{E}[\langle \nabla_{b_t} f(\boldsymbol{\theta}_t), \boldsymbol{\Delta}_t + \eta_l K \nabla_{b_t} f(\boldsymbol{\theta}_t) - \eta_l K \nabla_{b_t} f(\boldsymbol{\theta}_t) \rangle] \\
&= -\eta\eta_l K\mathbb{E}[\|\nabla_{b_t} f(\boldsymbol{\theta}_t)\|^2] + \eta\mathbb{E}[\langle \nabla_{b_t} f(\boldsymbol{\theta}_t), \boldsymbol{\Delta}_t + \eta_l K \nabla_{b_t} f(\boldsymbol{\theta}_t) \rangle].
\end{aligned} \tag{10}
$$

Then

$$
\begin{aligned}
&\eta\mathbb{E}[\langle \nabla_{b_t} f(\boldsymbol{\theta}_t), \boldsymbol{\Delta}_t + \eta_l K \nabla_{b_t} f(\boldsymbol{\theta}_t) \rangle] \\
&= \eta\mathbb{E}\Big[\Big\langle \nabla_{b_t} f(\boldsymbol{\theta}_t), \frac{1}{N}\sum_{i=1}^{N}\boldsymbol{\Delta}_t^i + \frac{\eta_l K}{N}\sum_{i=1}^{N}\nabla_{b_t} f(\boldsymbol{\theta}_t) \Big\rangle\Big] \\
&= \eta\mathbb{E}\Big[\Big\langle \nabla_{b_t} f(\boldsymbol{\theta}_t), -\frac{\eta_l}{N}\sum_{i=1}^{N}\sum_{k=0}^{K-1}\boldsymbol{g}_{t,k}^i + \frac{\eta_l K}{N}\sum_{i=1}^{N}\nabla_{b_t} f(\boldsymbol{\theta}_t) \Big\rangle\Big] \\
&= \eta\mathbb{E}\Big[\Big\langle \nabla_{b_t} f(\boldsymbol{\theta}_t), -\frac{\eta_l}{N}\sum_{i=1}^{N}\sum_{k=0}^{K-1}\nabla_{b_t} f_i(\boldsymbol{\theta}_{t,k}^i) + \frac{\eta_l K}{N}\sum_{i=1}^{N}\nabla_{b_t} f_i(\boldsymbol{\theta}_t) \Big\rangle\Big] \\
&= \eta\mathbb{E}\Big[\Big\langle \sqrt{\eta_l K} \cdot \nabla_{b_t} f(\boldsymbol{\theta}_t), -\frac{\sqrt{\eta_l K}}{NK}\sum_{i=1}^{N}\sum_{k=0}^{K-1}\nabla_{b_t} f_i(\boldsymbol{\theta}_{t,k}^i) + \frac{\sqrt{\eta_l K}}{N}\sum_{i=1}^{N}\nabla_{b_t} f_i(\boldsymbol{\theta}_t) \Big\rangle\Big] \\
&= \frac{\eta\eta_l K}{2}\mathbb{E}[\|\nabla_{b_t} f(\boldsymbol{\theta}_t)\|^2] + \frac{\eta\eta_l}{2N^2 K}\mathbb{E}\Big[\Big\|\sum_{i=1}^{N}\sum_{k=0}^{K-1}[\nabla_{b_t} f_i(\boldsymbol{\theta}_{t,k}^i) - \nabla_{b_t} f_i(\boldsymbol{\theta}_t)]\Big\|^2\Big] \\
&\quad - \frac{\eta\eta_l}{2N^2 K}\mathbb{E}\Big[\Big\|\sum_{i=1}^{N}\sum_{k=0}^{K-1}\nabla_{b_t} f_i(\boldsymbol{\theta}_{t,k}^i)\Big\|^2\Big],
\end{aligned} \tag{11}
$$

where the third equation holds by the unbiased-ness of stochastic gradient, and the last one holds by the fact of $\langle \boldsymbol{a}, \boldsymbol{b} \rangle = \frac{1}{2}[\|\boldsymbol{a}\|^2 + \|\boldsymbol{b}\|^2 + \|\boldsymbol{a} - \boldsymbol{b}\|^2]$. For the second term in Eq. (11), there is

$$
\begin{aligned}
\frac{\eta \eta_l}{2N^2 K} \mathbb{E} & \left[ \left\| \sum_{i=1}^{N} \sum_{k=0}^{K-1} [\nabla_{b_t} f_i(\boldsymbol{\theta}_{t,k}^i) - \nabla_{b_t} f_i(\boldsymbol{\theta}_t)] \right\|^2 \right] \\
& \leq \frac{\eta \eta_l}{2N} \sum_{i=1}^{N} \sum_{k=0}^{K-1} \mathbb{E}[\|\nabla_{b_t} f_i(\boldsymbol{\theta}_{t,k}^i) - \nabla_{b_t} f_i(\boldsymbol{\theta}_t)\|^2] \\
& \leq \frac{\eta \eta_l L^2}{2N} \sum_{i=1}^{N} \sum_{k=0}^{K-1} \mathbb{E}[\|\boldsymbol{\theta}_{t,k}^i - \boldsymbol{\theta}_t\|^2].
\end{aligned}
\tag{12}
$$

Therefore, for the whole $I_1$ term, we have

$$
\begin{aligned}
I_1 \leq {} & -\eta \eta_l K \mathbb{E}[\|\nabla_{b_t} f(\boldsymbol{\theta}_t)\|^2] + \frac{\eta \eta_l K}{2} \mathbb{E}[\|\nabla_{b_t} f(\boldsymbol{\theta}_t)\|^2] + \frac{\eta \eta_l L^2}{2N} \sum_{i=1}^{N} \sum_{k=0}^{K-1} \mathbb{E}[\|\boldsymbol{\theta}_{t,k}^i - \boldsymbol{\theta}_t\|^2] \\
& - \frac{\eta \eta_l}{2N^2 K} \mathbb{E} \left[ \left\| \sum_{i=1}^{N} \sum_{k=0}^{K-1} \nabla_{b_t} f_i(\boldsymbol{\theta}_{t,k}^i) \right\|^2 \right] \\
= {} & -\frac{\eta \eta_l K}{2} \mathbb{E}[\|\nabla_{b_t} f(\boldsymbol{\theta}_t)\|^2] + \frac{\eta \eta_l L^2}{2N} \sum_{i=1}^{N} \sum_{k=0}^{K-1} \mathbb{E}[\|\boldsymbol{\theta}_{t,k}^i - \boldsymbol{\theta}_t\|^2] - \frac{\eta \eta_l}{2N^2 K} \mathbb{E} \left[ \left\| \sum_{i=1}^{N} \sum_{k=0}^{K-1} \nabla_{b_t} f_i(\boldsymbol{\theta}_{t,k}^i) \right\|^2 \right].
\end{aligned}
\tag{13}
$$

where the equation holds by Lemma C.3.

Note that the model $\boldsymbol{\theta}_{t,k}^{i,b}$ is the $k$-th local step model for update block $b_t$ at time step $t$, thus the previous $b_{t-1}$ block has not been updated yet, i.e.,

$$
\begin{aligned}
\boldsymbol{\theta}_{t,k}^i &= \texttt{LocalBlockTraining}(\boldsymbol{\theta}_{t,0}^i, \eta_l, k), \\
\boldsymbol{\theta}_{t,0}^i = \boldsymbol{\theta}_t^i &= \boldsymbol{\theta}_{t-1}^i + \eta \bar{\boldsymbol{\Delta}}_{t-1}^i + \eta \bar{\boldsymbol{\Delta}}_{t-2} - \eta \bar{\boldsymbol{\Delta}}_{t-2}^i \\
&= \boldsymbol{\theta}_{t-1} + \eta \bar{\boldsymbol{\Delta}}_{t-1}^i,
\end{aligned}
$$

and for the global model,

$$
\boldsymbol{\theta}_t = \boldsymbol{\theta}_{t-1} + \eta \bar{\boldsymbol{\Delta}}_{t-1},
\tag{14}
$$

then

$$
\begin{aligned}
\mathbb{E}[\|\boldsymbol{\theta}_{t,k}^i - \boldsymbol{\theta}_t\|^2] &= \mathbb{E}[\|\boldsymbol{\theta}_{t,k}^i - \boldsymbol{\theta}_{t,0}^i + \boldsymbol{\theta}_{t,0}^i - \boldsymbol{\theta}_t\|^2] \\
&\leq 2\mathbb{E}[\|\boldsymbol{\theta}_{t,k}^i - \boldsymbol{\theta}_{t,0}^i\|^2] + 2\mathbb{E}[\|\boldsymbol{\theta}_{t,0}^i - \boldsymbol{\theta}_t\|^2],
\end{aligned}
\tag{15}
$$

where the first term consists of $k$ steps of local updates, while the second term includes the updates difference when updating the $b_{t-1}$ block. For $k = 0, \ldots, K-1$, we obtain

$$
\begin{aligned}
\mathbb{E}\big[\big\|\boldsymbol{\theta}_{t,k}^i - \boldsymbol{\theta}_{t,0}^i\big\|^2\big] &= \mathbb{E}\big[\big\|[\boldsymbol{\theta}_{t,k}^i - \boldsymbol{\theta}_{t,0}^i]_{b_t}\big\|^2\big] \\
&= \mathbb{E}\big[\big\|[\boldsymbol{\theta}_{t,k-1}^i - \boldsymbol{\theta}_{t,0}^i - \eta_l \boldsymbol{g}_{t,k}^i]_{b_t}\big\|^2\big] \\
&\leq \mathbb{E}\big[\big\|[\boldsymbol{\theta}_{t,k-1}^i]_{b_t} - [\boldsymbol{\theta}_{t,0}^i]_{b_t} - \eta_l([\boldsymbol{g}_{t,k}^i]_{b_t} - \nabla_{b_t} f_i(\boldsymbol{\theta}_{t,k-1}^i) + \nabla_{b_t} f_i(\boldsymbol{\theta}_{t,k-1}^i) - \nabla_{b_t} f_i(\boldsymbol{\theta}_{t,0}^i) + \nabla_{b_t} f_i(\boldsymbol{\theta}_{t,0}^i) \\
&\quad - \nabla_{b_t} f(\boldsymbol{\theta}_{t,0}^i) + \nabla_{b_t} f(\boldsymbol{\theta}_{t,0}^i))\big\|^2\big] \\
&\leq \left(1 + \frac{1}{2K-1}\right) \mathbb{E}\big[\big\|[\boldsymbol{\theta}_{t,k-1}^i]_{b_t} - [\boldsymbol{\theta}_{t,0}^i]_{b_t}\big\|^2\big] + \mathbb{E}\big[\big\|\eta_l([\boldsymbol{g}_{t,k}^i]_{b_t} - \nabla_{b_t} f_i(\boldsymbol{\theta}_{t,k-1}^i))\big\|^2\big] \\
&\quad + 6K\mathbb{E}[\|\eta_l(\nabla_{b_t} f_i(\boldsymbol{\theta}_{t,k-1}^i) - \nabla_{b_t} f_i(\boldsymbol{\theta}_{t,0}^i))\|^2] + 6K\mathbb{E}[\|\eta_l(\nabla_{b_t} f_i(\boldsymbol{\theta}_{t,0}^i) - \nabla_{b_t} f(\boldsymbol{\theta}_{t,0}^i))\|^2] \\
&\quad + 6K\mathbb{E}[\|\eta_l \nabla_{b_t} f(\boldsymbol{\theta}_{t,0}^i)\|^2] \\
&\leq \left(1 + \frac{1}{2K-1} + 6K\eta_l^2 L^2\right) \mathbb{E}\big[\big\|[\boldsymbol{\theta}_{t,k-1}^i]_{b_t} - [\boldsymbol{\theta}_{t,0}^i]_{b_t}\big\|^2\big] + \eta_l^2 \sigma^2 \\
&\quad + 6K\mathbb{E}[\|\eta_l(\nabla_{b_t} f_i(\boldsymbol{\theta}_{t,0}^i) - \nabla_{b_t} f(\boldsymbol{\theta}_{t,0}^i))\|^2] + 6K\mathbb{E}[\|\eta_l \nabla_{b_t} f(\boldsymbol{\theta}_{t,0}^i)\|^2] \\
&= \left(1 + \frac{1}{2K-1} + 6K\eta_l^2 L^2\right) \mathbb{E}[\|\boldsymbol{\theta}_{t,k-1}^i - \boldsymbol{\theta}_{t,0}^i\|^2] + \eta_l^2 \sigma^2 \\
&\quad + 6K\mathbb{E}[\|\eta_l(\nabla_{b_t} f_i(\boldsymbol{\theta}_{t,0}^i) - \nabla_{b_t} f(\boldsymbol{\theta}_{t,0}^i))\|^2] + 6K\mathbb{E}[\|\eta_l \nabla_{b_t} f(\boldsymbol{\theta}_{t,0}^i)\|^2],
\end{aligned}
\tag{16}
$$

then by taking average over all clients $i \in [N]$,

$$
\begin{aligned}
\frac{1}{N}\sum_{i=1}^N \mathbb{E}[\|\boldsymbol{\theta}_{t,k}^i - \boldsymbol{\theta}_{t,0}^i\|^2] \leq &\ \frac{1}{N}\sum_{i=1}^N \left(1 + \frac{1}{2K-1} + 6K\eta_l^2 L^2\right) \mathbb{E}[\|\boldsymbol{\theta}_{t,k-1}^i - \boldsymbol{\theta}_{t,0}^i\|^2] + \eta_l^2 \sigma^2 \\
&+ 6K\eta_l^2 \sigma_g^2 + \frac{6K\eta_l^2}{N}\sum_{i=1}^N \mathbb{E}[\|\nabla_{b_t} f(\boldsymbol{\theta}_{t,0}^i)\|^2].
\end{aligned}
\tag{17}
$$

Since $\eta_l \leq \frac{1}{8KL}$, unrolling the recursion, then we have

$$
\begin{aligned}
\frac{1}{N}&\sum_{i=1}^N \mathbb{E}[\|\boldsymbol{\theta}_{t,k}^i - \boldsymbol{\theta}_{t,0}^i\|^2] \\
&\leq \sum_{p=0}^{k-1} \left(1 + \frac{1}{K-1}\right)^p \left[\eta_l^2 \sigma^2 + 6K\eta_l^2 \sigma_g^2 + \frac{6K\eta_l^2}{N}\sum_{i=1}^N \mathbb{E}[\|\nabla_{b_t} f(\boldsymbol{\theta}_{t,0}^i)\|^2]\right] \\
&\leq (K-1)\left[\left(1 + \frac{1}{K-1}\right)^K - 1\right]\left[\eta_l^2 \sigma^2 + 6K\eta_l^2 \sigma_g^2 + \frac{6K\eta_l^2}{N}\sum_{i=1}^N \mathbb{E}[\|\nabla_{b_t} f(\boldsymbol{\theta}_{t,0}^i)\|^2]\right] \\
&\leq 5K\eta_l^2 \sigma^2 + 30K^2\eta_l^2 \sigma_g^2 + \frac{30K^2\eta_l^2}{N}\sum_{i=1}^N \mathbb{E}[\|\nabla_{b_t} f(\boldsymbol{\theta}_{t,0}^i)\|^2],
\end{aligned}
\tag{18}
$$

for the last item, we have

$$
\begin{aligned}
\frac{1}{N}\sum_{i=1}^N \mathbb{E}[\|\nabla_{b_t} f(\boldsymbol{\theta}_{t,0}^i)\|^2] &= \frac{1}{N}\sum_{i=1}^N \mathbb{E}[\|\nabla_{b_t} f(\boldsymbol{\theta}_{t,0}^i) - \nabla_{b_t} f(\boldsymbol{\theta}_t) + \nabla_{b_t} f(\boldsymbol{\theta}_t)\|^2] \\
&\leq \frac{2}{N}\sum_{i=1}^N \mathbb{E}[\|\nabla_{b_t} f(\boldsymbol{\theta}_{t,0}^i) - \nabla_{b_t} f(\boldsymbol{\theta}_t)\|^2] + \frac{2}{N}\sum_{i=1}^N \mathbb{E}[\|\nabla_{b_t} f(\boldsymbol{\theta}_t)\|^2] \\
&\leq \frac{2L^2}{N}\sum_{i=1}^N \mathbb{E}[\|\boldsymbol{\theta}_{t,0}^i - \boldsymbol{\theta}_t\|^2] + 2\mathbb{E}[\|\nabla_{b_t} f(\boldsymbol{\theta}_t)\|^2],
\end{aligned}
\tag{19}
$$

where there is

$$
\begin{aligned}
\mathbb{E}[\|\boldsymbol{\theta}_{t,0}^i - \boldsymbol{\theta}_t\|^2] &= \mathbb{E}[\|\boldsymbol{\theta}_{t-1} + \eta\bar{\boldsymbol{\Delta}}_{t-1}^i - (\boldsymbol{\theta}_{t-1} + \eta\bar{\boldsymbol{\Delta}}_{t-1})\|^2] \\
&= \eta^2\mathbb{E}[\|\bar{\boldsymbol{\Delta}}_{t-1}^i - \bar{\boldsymbol{\Delta}}_{t-1}\|^2] \\
&\leq 2\eta^2\mathbb{E}[\|\bar{\boldsymbol{\Delta}}_{t-1}^i\|^2] + 2\eta^2\mathbb{E}[\|\bar{\boldsymbol{\Delta}}_{t-1}\|^2] \\
&= 2\eta^2\mathbb{E}[\|\boldsymbol{\Delta}_{t-1}^i\|^2] + 2\eta^2\mathbb{E}[\|\boldsymbol{\Delta}_{t-1}\|^2].
\end{aligned}
\tag{20}
$$

Merging items together, then we obtain

$$
\begin{aligned}
\frac{1}{N}\sum_{i=1}^N \mathbb{E}[\|\boldsymbol{\theta}_{t,k}^i - \boldsymbol{\theta}_t\|^2] &\leq \frac{2}{N}\sum_{i=1}^N \mathbb{E}[\|\boldsymbol{\theta}_{t,k}^i - \boldsymbol{\theta}_{t,0}^i\|^2] + \frac{2}{N}\sum_{i=1}^N \mathbb{E}[\|\boldsymbol{\theta}_{t,0}^i - \boldsymbol{\theta}_t\|^2] \\
&\leq 10K\eta_l^2\sigma^2 + 60K^2\eta_l^2\sigma_g^2 + \frac{60K^2\eta_l^2}{N}\sum_{i=1}^N \mathbb{E}[\|\nabla_{b_t}f(\boldsymbol{\theta}_{t,0}^i)\|^2] + \frac{2}{N}\sum_{i=1}^N \mathbb{E}[\|\boldsymbol{\theta}_{t,0}^i - \boldsymbol{\theta}_t\|^2] \\
&\leq 10K\eta_l^2\sigma^2 + 60K^2\eta_l^2\sigma_g^2 + 120K^2\eta_l^2\mathbb{E}[\|\nabla_{b_t}f(\boldsymbol{\theta}_t)\|^2] \\
&\quad + \frac{120K^2\eta_l^2 L^2}{N}\sum_{i=1}^N \mathbb{E}[\|\boldsymbol{\theta}_{t,0}^i - \boldsymbol{\theta}_t\|^2] + \frac{2}{N}\sum_{i=1}^N \mathbb{E}[\|\boldsymbol{\theta}_{t,0}^i - \boldsymbol{\theta}_t\|^2] \\
&\leq 10K\eta_l^2\sigma^2 + 60K^2\eta_l^2\sigma_g^2 + 120K^2\eta_l^2\mathbb{E}[\|\nabla_{b_t}f(\boldsymbol{\theta}_t)\|^2] + \frac{4}{N}\sum_{i=1}^N \mathbb{E}[\|\boldsymbol{\theta}_{t,0}^i - \boldsymbol{\theta}_t\|^2] \\
&\leq 10K\eta_l^2\sigma^2 + 60K^2\eta_l^2\sigma_g^2 + 120K^2\eta_l^2\mathbb{E}[\|\nabla_{b_t}f(\boldsymbol{\theta}_t)\|^2] + 8\eta^2\mathbb{E}[\|\boldsymbol{\Delta}_{t-1}\|^2] + \frac{8\eta^2}{N}\sum_{i=1}^N \mathbb{E}[\|\boldsymbol{\Delta}_{t-1}^i\|^2].
\end{aligned}
\tag{21}
$$

Therefore, reorganizing the $I_1$ term, we obtain

$$
\begin{aligned}
I_1 &\leq -\frac{\eta\eta_l K}{2}\mathbb{E}[\|\nabla_{b_t}f(\boldsymbol{\theta}_t)\|^2] + \frac{\eta\eta_l L^2}{2N}\sum_{i=1}^N\sum_{k=0}^{K-1}\mathbb{E}[\|\boldsymbol{\theta}_{t,k}^i - \boldsymbol{\theta}_t\|^2] - \frac{\eta\eta_l}{2N^2 K}\mathbb{E}\left[\left\|\sum_{i=1}^N\sum_{k=0}^{K-1}\nabla_{b_t}f_i(\boldsymbol{\theta}_{t,k}^i)\right\|^2\right] \\
&\leq -\frac{\eta\eta_l K}{2}\mathbb{E}[\|\nabla_{b_t}f(\boldsymbol{\theta}_t)\|^2] + \eta\eta_l L^2 K\left[5K\eta_l^2\sigma^2 + 30K^2\eta_l^2\sigma_g^2 + 60K^2\eta_l^2\mathbb{E}[\|\nabla_{b_t}f(\boldsymbol{\theta}_t)\|^2] \right. \\
&\quad \left. + 4\eta^2\mathbb{E}[\|\boldsymbol{\Delta}_{t-1}\|^2] + \frac{4\eta^2}{N}\sum_{i=1}^N \mathbb{E}[\|\boldsymbol{\Delta}_{t-1}^i\|^2]\right] - \frac{\eta\eta_l}{2N^2 K}\mathbb{E}\left[\left\|\sum_{i=1}^N\sum_{k=0}^{K-1}\nabla_{b_t}f_i(\boldsymbol{\theta}_{t,k}^i)\right\|^2\right]
\end{aligned}
\tag{22}
$$

Summing up Eq. (9),

$$
\begin{aligned}
\mathbb{E}[f(\boldsymbol{\theta}_T)] - f(\boldsymbol{\theta}_0) &\leq \eta \sum_{t=0}^{T-1} \mathbb{E}[\langle \nabla_{b_t} f(\boldsymbol{\theta}_T), \boldsymbol{\Delta}_t \rangle] + \frac{\eta^2 L}{2} \sum_{t=0}^{T-1} \mathbb{E}[\|\boldsymbol{\Delta}_t\|^2] \\
&\leq -\frac{\eta \eta_l K}{2} \sum_{t=0}^{T-1} \mathbb{E}[\|\nabla_{b_t} f(\boldsymbol{\theta}_t)\|^2] + \eta \eta_l L^2 K \Bigg[ 5TK\eta_l^2 \sigma^2 + 30TK^2 \eta_l^2 \sigma_g^2 \\
&\quad + 60K^2 \eta_l^2 \sum_{t=0}^{T-1} \mathbb{E}[\|\nabla_{b_t} f(\boldsymbol{\theta}_t)\|^2] + 4\eta^2 \sum_{t=0}^{T-1} \mathbb{E}[\|\boldsymbol{\Delta}_{t-1}\|^2] + \frac{4\eta^2}{N} \sum_{t=0}^{T-1} \sum_{i=1}^{N} \mathbb{E}[\|\boldsymbol{\Delta}_{t-1}^i\|^2] \Bigg] \\
&\quad - \frac{\eta \eta_l}{2N^2 K} \sum_{t=0}^{T-1} \mathbb{E}\Bigg[ \Bigg\| \sum_{i=1}^{N} \sum_{k=0}^{K-1} \nabla_{b_t} f_i(\boldsymbol{\theta}_{t,k}^i) \Bigg\|^2 \Bigg] + \frac{\eta^2 L}{2} \sum_{t=0}^{T-1} \mathbb{E}[\|\boldsymbol{\Delta}_t\|^2] \\
&\leq -\frac{\eta \eta_l K}{2} \sum_{t=0}^{T-1} \mathbb{E}[\|\nabla_{b_t} f(\boldsymbol{\theta}_t)\|^2] + 60K^3 \eta \eta_l^3 L^2 \sum_{t=0}^{T-1} \mathbb{E}[\|\nabla_{b_t} f(\boldsymbol{\theta}_t)\|^2] \\
&\quad + \eta \eta_l L^2 K (5TK\eta_l^2 \sigma^2 + 30TK^2 \eta_l^2 \sigma_g^2) + \left( 4\eta^3 \eta_l L^2 K + \frac{\eta^2 L}{2} \right) \sum_{t=0}^{T} \mathbb{E}[\|\boldsymbol{\Delta}_t\|^2] \\
&\quad + \frac{4\eta^3 \eta_l L^2 K}{N} \sum_{t=0}^{T-1} \sum_{i=1}^{N} \mathbb{E}[\|\boldsymbol{\Delta}_{t-1}^i\|^2] - \frac{\eta \eta_l}{2N^2 K} \sum_{t=0}^{T-1} \mathbb{E}\Bigg[ \Bigg\| \sum_{i=1}^{N} \sum_{k=0}^{K-1} \nabla_{b_t} f_i(\boldsymbol{\theta}_{t,k}^i) \Bigg\|^2 \Bigg]. \quad (23)
\end{aligned}
$$

By Lemma C.2, the inequality becomes

$$
\begin{aligned}
\mathbb{E}[f(\boldsymbol{\theta}_T)] - f(\boldsymbol{\theta}_0) &\leq -\frac{\eta \eta_l K}{2} \sum_{t=0}^{T-1} \mathbb{E}[\|\nabla_{b_t} f(\boldsymbol{\theta}_t)\|^2] + 60K^3 \eta \eta_l^3 L^2 \sum_{t=0}^{T-1} \mathbb{E}[\|\nabla_{b_t} f(\boldsymbol{\theta}_t)\|^2] \\
&\quad + \eta \eta_l L^2 K (5TK\eta_l^2 \sigma^2 + 30TK^2 \eta_l^2 \sigma_g^2) + \left( 4\eta^3 \eta_l L^2 K + \frac{\eta^2 L}{2} \right) \sum_{t=0}^{T-1} \mathbb{E}[\|\boldsymbol{\Delta}_t\|^2] \\
&\quad + 4\eta^3 \eta_l L^2 K \sum_{t=0}^{T-1} \mathbb{E}[\|\boldsymbol{\Delta}_t\|^2] + \frac{\eta \eta_l K}{4} \sum_{t=0}^{T-1} \mathbb{E}[\|\nabla_{b_t} f(\boldsymbol{\theta}_t)\|^2] \\
&\quad + 4\eta^3 \eta_l L^2 K (2TK\eta_l^2 \sigma^2 + 20TK^3 \eta_l^4 L^2 \sigma^2 + 120TK^4 \eta_l^4 L^2 \sigma_g^2) \\
&\quad - \frac{\eta \eta_l}{2N^2 K} \sum_{t=0}^{T-1} \mathbb{E}\Bigg[ \Bigg\| \sum_{i=1}^{N} \sum_{k=0}^{K-1} \nabla_{b_t} f_i(\boldsymbol{\theta}_{t,k}^i) \Bigg\|^2 \Bigg]. \quad (24)
\end{aligned}
$$

By condition on learning rates, i.e., $\eta_l \leq \frac{1}{22KL}$ and $\eta_l \leq \frac{1}{4KL\eta}$,

$$
\begin{aligned}
\mathbb{E}[f(\boldsymbol{\theta}_T)] - f(\boldsymbol{\theta}_0) \leq & -\frac{\eta\eta_l K}{8}\sum_{t=0}^{T-1}\mathbb{E}[\|\nabla_{b_t}f(\boldsymbol{\theta}_t)\|^2] + \eta\eta_l L^2 K(5TK\eta_l^2\sigma^2 + 30TK^2\eta_l^2\sigma_g^2) \\
& + \left(8\eta^3\eta_l L^2 K + \frac{\eta^2 L}{2}\right)\sum_{t=0}^{T-1}\mathbb{E}[\|\boldsymbol{\Delta}_t\|^2] \\
& + 4\eta^3\eta_l L^2 K(2TK\eta_l^2\sigma^2 + 20TK^3\eta_l^4 L^2\sigma^2 + 120TK^4\eta_l^4 L^2\sigma_g^2) \\
& - \frac{\eta\eta_l}{2N^2 K}\sum_{t=0}^{T-1}\mathbb{E}\left[\left\|\sum_{i=1}^{N}\sum_{k=0}^{K-1}\nabla_{b_t}f_i(\boldsymbol{\theta}_{t,k}^i)\right\|^2\right] \\
\leq & -\frac{\eta\eta_l K}{8}\sum_{t=0}^{T-1}\mathbb{E}[\|\nabla_{b_t}f(\boldsymbol{\theta}_t)\|^2] + \eta\eta_l L^2 K(5TK\eta_l^2\sigma^2 + 30TK^2\eta_l^2\sigma_g^2) \\
& + \left(8\eta^3\eta_l L^2 K + \frac{\eta^2 L}{2}\right)\frac{TK\eta_l^2}{N}\sigma^2 \\
& + 4\eta^3\eta_l L^2 K(2TK\eta_l^2\sigma^2 + 20TK^3\eta_l^4 L^2\sigma^2 + 120TK^4\eta_l^4 L^2\sigma_g^2).
\end{aligned}
\tag{25}
$$

Then,

$$
\begin{aligned}
\sum_{t=0}^{T-1}\mathbb{E}[\|\nabla_{b_t}f(\boldsymbol{\theta}_t)\|^2] \leq & \frac{8}{\eta\eta_l K}\left[f(\boldsymbol{\theta}_0) - \mathbb{E}[f(\boldsymbol{\theta}_T)]\right] + 40TK\eta_l^2 L^2\sigma^2 + 240TK^2\eta_l^2 L^2\sigma_g^2 \\
& + \left(8\eta^2\eta_l L^2 K + \frac{\eta L}{2}\right)\frac{T\eta_l}{N}\sigma^2 \\
& + 32\eta^2 L^2(2TK\eta_l^2\sigma^2 + 20TK^3\eta_l^4 L^2\sigma^2 + 120TK^4\eta_l^4 L^2\sigma_g^2).
\end{aligned}
\tag{26}
$$

Dividing by $T$, there is

$$
\begin{aligned}
\frac{1}{T}\sum_{t=0}^{T-1}\mathbb{E}[\|\nabla_{b_t}f(\boldsymbol{\theta}_t)\|^2] \leq & \frac{8}{\eta\eta_l TK}\left[f(\boldsymbol{\theta}_0) - \mathbb{E}[f(\boldsymbol{\theta}_T)]\right] + 40K\eta_l^2 L^2\sigma^2 + 240K^2\eta_l^2 L^2\sigma_g^2 \\
& + \left(8\eta^2\eta_l L^2 K + \frac{\eta L}{2}\right)\frac{\eta_l}{N}\sigma^2 \\
& + 32\eta^2 L^2(2K\eta_l^2\sigma^2 + 20K^3\eta_l^4 L^2\sigma^2 + 120K^4\eta_l^4 L^2\sigma_g^2).
\end{aligned}
\tag{27}
$$

With $\mathcal{F} = f(\boldsymbol{\theta}_0) - f_*$ and $f_* = \min_{\boldsymbol{\theta}}f(\boldsymbol{\theta}) > -\infty$, and there is $f(\boldsymbol{\theta}_0) - \mathbb{E}[f(\boldsymbol{\theta}_T)] \leq f(\boldsymbol{\theta}_0) - f_* = \mathcal{F}$, then

$$
\begin{aligned}
\frac{1}{T}\sum_{t=0}^{T-1}\mathbb{E}[\|\nabla_{b_t}f(\boldsymbol{\theta}_t)\|^2] \leq & \frac{8}{\eta\eta_l TK}\left[f(\boldsymbol{\theta}_0) - \mathbb{E}[f(\boldsymbol{\theta}_T)]\right] + 40K\eta_l^2 L^2\sigma^2 + 240K^2\eta_l^2 L^2\sigma_g^2 \\
& + \left(8\eta^2\eta_l L^2 K + \frac{\eta L}{2}\right)\frac{\eta_l}{N}\sigma^2 \\
& + 32\eta^2 L^2(2K\eta_l^2\sigma^2 + 20K^3\eta_l^4 L^2\sigma^2 + 120K^4\eta_l^4 L^2\sigma_g^2) \\
\leq & \frac{8}{\eta\eta_l TK}\mathcal{F} + 40K\eta_l^2 L^2\sigma^2 + 240K^2\eta_l^2 L^2\sigma_g^2 \\
& + \left(8\eta^2\eta_l L^2 K + \frac{\eta L}{2}\right)\frac{\eta_l}{N}\sigma^2 \\
& + 32\eta^2 L^2(2K\eta_l^2\sigma^2 + 20K^3\eta_l^4 L^2\sigma^2 + 120K^4\eta_l^4 L^2\sigma_g^2) \\
= & \frac{8\mathcal{F}}{\eta\eta_l TK} + 40\eta_l^2 L^2 K(\sigma^2 + 6K\sigma_g^2) \\
& + \left(8\eta^2\eta_l L^2 K + \frac{\eta L}{2}\right)\frac{\eta_l}{N}\sigma^2 + 64\eta_l^2\eta^2 L^2 K[\sigma^2 + 10\eta_l^2 L^2 K^2(\sigma^2 + 6K\sigma_g^2)].
\end{aligned}
\tag{28}
$$

This concludes the proof of Theorem 5.3.

## B.3 Extension to Local Adaptive Optimizer

The theoretical analysis of the proposed ParaBlock is not limited to the local SGD setting. Essentially, the main differences between the convergence analysis under SGD and adaptive optimizers can be summarized as follows:

- The local updates $\boldsymbol{\Delta}_t^i$ are aggregated to $\boldsymbol{\Delta}_t$ on the server. Hence, the most crucial part of modifying to AdamW is to deal with these $\boldsymbol{\Delta}$ terms.

- For $\boldsymbol{\Delta}_t^i$ in Adam, there is $\boldsymbol{\Delta}_t^i = \boldsymbol{\theta}_{t,K}^i - \boldsymbol{\theta}_{t,0}^i = \sum_{k=1}^{K}(\boldsymbol{\theta}_{t,k}^i - \boldsymbol{\theta}_{t,k-1}^i)$. Thus there is

$$
\begin{aligned}
\boldsymbol{\Delta}_t &= \frac{1}{N}\sum_{i=1}^{N}\boldsymbol{\Delta}_t^i = \frac{1}{N}\sum_{i=1}^{N}[\boldsymbol{\theta}_{t,K}^i - \boldsymbol{\theta}_{t,0}^i] = \frac{1}{N}\sum_{i=1}^{N}\sum_{k=1}^{K}(\boldsymbol{\theta}_{t,k}^i - \boldsymbol{\theta}_{t,k-1}^i) \\
&= \frac{1}{N}\sum_{i=1}^{N}\sum_{k=1}^{K}\eta_l\frac{\boldsymbol{m}_{t,k}^i}{\sqrt{\boldsymbol{v}_{t,k}^i}+\epsilon}, \\
\Rightarrow \|\boldsymbol{\Delta}_t\|^2 &= \left\|\frac{1}{N}\sum_{i=1}^{N}\sum_{k=1}^{K}\eta_l\frac{\boldsymbol{m}_{t,k}^i}{\sqrt{\boldsymbol{v}_{t,k}^i}+\epsilon}\right\|^2 \\
&\le \frac{\eta_l^2}{\epsilon}\left\|\frac{1}{N}\sum_{i=1}^{N}\sum_{k=1}^{K}\boldsymbol{m}_{t,k}^i\right\|^2 \\
&= \frac{\eta_l^2}{\epsilon}\left\|\frac{1}{N}\sum_{i=1}^{N}\sum_{k=1}^{K}\sum_{j=1}^{k}(1-\beta_1)\beta_1^{k-j}\boldsymbol{g}_{t,j}^i\right\|^2 \\
&= \frac{\eta_l^2}{\epsilon}\left\|\frac{1}{N}\sum_{i=1}^{N}\sum_{k=1}^{K}(1-\beta_1^{K-k+1})\boldsymbol{g}_{t,k}^i\right\|^2,
\end{aligned}
\tag{29}
$$

therefore, by similar theoretical analysis in Lemma C.1, we have

$$
\begin{aligned}
\mathbb{E}[\|\boldsymbol{\Delta}_t\|^2] &= \frac{\eta_l^2}{\epsilon}\mathbb{E}\left[\left\|\frac{1}{N}\sum_{i=1}^{N}\sum_{k=0}^{K-1}(1-\beta_1^{K-k+1})[\boldsymbol{g}_{t,k}^i - \nabla_{b_t}f_i(\boldsymbol{\theta}_{t,k}^i)+\nabla_{b_t}f_i(\boldsymbol{\theta}_{t,k}^i)]\right\|^2\right] \\
&= \frac{\eta_l^2}{\epsilon}\mathbb{E}\left[\left\|\frac{1}{N}\sum_{i=1}^{N}\sum_{k=0}^{K-1}(1-\beta_1^{K-k+1})[\boldsymbol{g}_{t,k}^i - \nabla_{b_t}f_i(\boldsymbol{\theta}_{t,k}^i)]\right\|^2\right] \\
&\quad + \frac{\eta_l^2}{\epsilon}\mathbb{E}\left[\left\|\frac{1}{N}\sum_{i=1}^{N}\sum_{k=0}^{K-1}(1-\beta_1^{K-k+1})\nabla_{b_t}f_i(\boldsymbol{\theta}_{t,k}^i)\right\|^2\right] \\
&\le \frac{K\eta_l^2}{N\epsilon}\sigma^2 + \frac{\eta_l^2}{N^2\epsilon^2}\left[\left\|\sum_{i=1}^{N}\sum_{k=0}^{K-1}\nabla_{b_t}f_i(\boldsymbol{\theta}_{t,k}^i)\right\|^2\right].
\end{aligned}
\tag{30}
$$

- The properties about bounding $\sum_{t=0}^{T-1}\frac{1}{N}\sum_{i=1}^{N}\mathbb{E}[\|\boldsymbol{\Delta}_t^i\|^2]$ would be also similar to the analysis in Lemma C.2.

- In a nutshell, adopting local Adam achieves the same convergence rate of $O(1/\sqrt{T})$ as SGD.

## C Supporting Lemmas

**Lemma C.1.** *The global update parameter* $\boldsymbol{\Delta}_t = \frac{1}{N} \sum_{i=1}^{N} \boldsymbol{\Delta}_t^i$ *satisfies*

$$\mathbb{E}[\|\boldsymbol{\Delta}_t\|^2] \leq \frac{K\eta_l^2}{N}\sigma^2 + \frac{\eta_l^2}{N^2}\mathbb{E}\left[\left\|\sum_{i=1}^{N}\sum_{k=0}^{K-1}\nabla_{b_t} f_i(\boldsymbol{\theta}_{t,k}^i)\right\|^2\right]. \tag{31}$$

*Proof.* By definition,

$$\begin{aligned}
\mathbb{E}[\|\boldsymbol{\Delta}_t\|^2] &= \mathbb{E}\left[\left\| - \frac{\eta_l}{N}\sum_{i=1}^{N}\sum_{k=0}^{K-1}\boldsymbol{g}_{t,k}^i\right\|^2\right] \\
&= \mathbb{E}\left[\left\| - \frac{\eta_l}{N}\sum_{i=1}^{N}\sum_{k=0}^{K-1}[\boldsymbol{g}_{t,k}^i - \nabla_{b_t} f_i(\boldsymbol{\theta}_{t,k}^i) + \nabla_{b_t} f_i(\boldsymbol{\theta}_{t,k}^i)]\right\|^2\right] \\
&= \mathbb{E}\left[\left\|\frac{\eta_l}{N}\sum_{i=1}^{N}\sum_{k=0}^{K-1}[\boldsymbol{g}_{t,k}^i - \nabla_{b_t} f_i(\boldsymbol{\theta}_{t,k}^i)]\right\|^2\right] + \mathbb{E}\left[\left\|\frac{\eta_l}{N}\sum_{i=1}^{N}\sum_{k=0}^{K-1}\nabla_{b_t} f_i(\boldsymbol{\theta}_{t,k}^i)\right\|^2\right] \\
&\leq \frac{K\eta_l^2}{N}\sigma^2 + \frac{\eta_l^2}{N^2}\mathbb{E}\left[\left\|\sum_{i=1}^{N}\sum_{k=0}^{K-1}\nabla_{b_t} f_i(\boldsymbol{\theta}_{t,k}^i)\right\|^2\right],
\end{aligned} \tag{32}$$

where the third equation holds by the unbiased-ness of the stochastic gradient, and the inequality holds by Assumption 5.2. $\square$

**Lemma C.2.** *The global update parameter* $\boldsymbol{\Delta}_t^i$ *satisfies*

$$\begin{aligned}
\sum_{t=0}^{T-1}\frac{1}{N}\sum_{i=1}^{N}\mathbb{E}[\|\boldsymbol{\Delta}_t^i\|^2] \leq &\sum_{t=0}^{T-1}\mathbb{E}[\|\boldsymbol{\Delta}_t\|^2] + \frac{1}{2\eta^2 L^2}\sum_{t=0}^{T-1}\mathbb{E}[\|\nabla_{b_t} f(\boldsymbol{\theta}_t)\|^2] \\
&+ 2TK\eta_l^2\sigma^2 + 20TK^2\eta_l^4 L^2\sigma^2 + 120TK^3\eta_l^4 L^2\sigma_g^2.
\end{aligned} \tag{33}$$

*Proof.* By definition,

$$\begin{aligned}
\mathbb{E}[\|\boldsymbol{\Delta}_t^i\|^2] &= \mathbb{E}\left[\left\| - \eta_l\sum_{k=0}^{K-1}\boldsymbol{g}_{t,k}^i\right\|^2\right] \\
&= \mathbb{E}\left[\left\| - \eta_l\sum_{k=0}^{K-1}[\boldsymbol{g}_{t,k}^i - \nabla_{b_t} f_i(\boldsymbol{\theta}_{t,k}^i) + \nabla_{b_t} f_i(\boldsymbol{\theta}_{t,k}^i)]\right\|^2\right] \\
&= \mathbb{E}\left[\left\|\eta_l\sum_{k=0}^{K-1}[\boldsymbol{g}_{t,k}^i - \nabla_{b_t} f_i(\boldsymbol{\theta}_{t,k}^i)]\right\|^2\right] + \mathbb{E}\left[\left\|\eta_l\sum_{k=0}^{K-1}\nabla_{b_t} f_i(\boldsymbol{\theta}_{t,k}^i)\right\|^2\right] \\
&\leq K\eta_l^2\sigma^2 + \eta_l^2\mathbb{E}\left[\left\|\sum_{k=0}^{K-1}\nabla_{b_t} f_i(\boldsymbol{\theta}_{t,k}^i)\right\|^2\right],
\end{aligned} \tag{34}$$

where the third equation holds by the unbiased-ness of the stochastic gradient, and the inequality holds by Assumption 5.2.

$$\mathbb{E}\left[\left\|\sum_{k=0}^{K-1}\nabla_{b_t}f_i(\boldsymbol{\theta}_{t,k}^i)\right\|^2\right]$$

$$= \mathbb{E}\left[\left\|\sum_{k=0}^{K-1}[\nabla_{b_t}f_i(\boldsymbol{\theta}_{t,k}^i)-\nabla_{b_t}f_i(\boldsymbol{\theta}_{t,0}^i)+\nabla_{b_t}f_i(\boldsymbol{\theta}_{t,0}^i)-\nabla_{b_t}f_i(\boldsymbol{\theta}_t)+\nabla_{b_t}f_i(\boldsymbol{\theta}_t)-\nabla_{b_t}f(\boldsymbol{\theta}_t)+\nabla_{b_t}f_i(\boldsymbol{\theta}_t)]\right\|^2\right]$$

$$\leq 2\mathbb{E}\left[\left\|\sum_{k=0}^{K-1}[\nabla_{b_t}f_i(\boldsymbol{\theta}_{t,k}^i)-\nabla_{b_t}f_i(\boldsymbol{\theta}_{t,0}^i)]\right\|^2\right] + 2\mathbb{E}\left[\left\|\sum_{k=0}^{K-1}\nabla_{b_t}f_i(\boldsymbol{\theta}_{t,0}^i)\right\|^2\right]$$

$$\leq 2K\sum_{k=0}^{K-1}L^2\mathbb{E}\left[\left\|\boldsymbol{\theta}_{t,k}^i-\boldsymbol{\theta}_{t,0}^i\right\|^2\right] + 2K^2\mathbb{E}\left[\left\|\nabla_{b_t}f_i(\boldsymbol{\theta}_{t,0}^i)\right\|^2\right], \tag{35}$$

where

$$\frac{1}{N}\sum_{i=1}^N\mathbb{E}[\|\boldsymbol{\theta}_{t,k}^i-\boldsymbol{\theta}_{t,0}^i\|^2]\leq 5K\eta_l^2\sigma^2+30K^2\eta_l^2\sigma_g^2+\frac{30K^2\eta_l^2}{N}\sum_{i=1}^N\mathbb{E}[\|\nabla_{b_t}f(\boldsymbol{\theta}_{t,0}^i)\|^2], \tag{36}$$

and

$$\frac{1}{N}\sum_{i=1}^N\mathbb{E}[\|\nabla_{b_t}f(\boldsymbol{\theta}_{t,0}^i)\|^2]\leq \frac{2L^2}{N}\sum_{i=1}^N\mathbb{E}[\|\boldsymbol{\theta}_{t,0}^i-\boldsymbol{\theta}_t\|^2]+2\mathbb{E}[\|\nabla_{b_t}f(\boldsymbol{\theta}_t)\|^2], \tag{37}$$

where there is

$$\mathbb{E}[\|\boldsymbol{\theta}_{t,0}^i-\boldsymbol{\theta}_t\|^2]\leq 2\eta^2\mathbb{E}[\|\boldsymbol{\Delta}_{t-1}^i\|^2]+2\eta^2\mathbb{E}[\|\boldsymbol{\Delta}_{t-1}\|^2]. \tag{38}$$

Then

$$\frac{1}{N}\sum_{i=1}^N\mathbb{E}[\|\boldsymbol{\Delta}_t^i\|^2]\leq K\eta_l^2\sigma^2+\frac{\eta_l^2}{N}\sum_{i=1}^N\mathbb{E}\left[\left\|\sum_{k=0}^{K-1}\nabla_{b_t}f_i(\boldsymbol{\theta}_{t,k}^i)\right\|^2\right]$$

$$\leq K\eta_l^2\sigma^2+\frac{2K\eta_l^2L^2}{N}\sum_{i=1}^N\sum_{k=0}^{K-1}\mathbb{E}[\|\boldsymbol{\theta}_{t,k}^i-\boldsymbol{\theta}_{t,0}^i\|^2]+\frac{2K^2\eta_l^2}{N}\sum_{i=1}^N\mathbb{E}[\|\nabla_{b_t}f(\boldsymbol{\theta}_{t,0}^i)\|^2]$$

$$\leq K\eta_l^2\sigma^2+10K^3\eta_l^4L^2\sigma^2+60K^4\eta_l^4L^2\sigma_g^2+\left(\frac{60K^4\eta_l^4L^2}{N}+\frac{2K^2\eta_l^2}{N}\right)\sum_{i=1}^N\mathbb{E}[\|\nabla_{b_t}f(\boldsymbol{\theta}_{t,0}^i)\|^2]$$

$$\leq K\eta_l^2\sigma^2+10K^3\eta_l^4L^2\sigma^2+60K^4\eta_l^4L^2\sigma_g^2+\left(\frac{120K^4\eta_l^4L^4}{N}+\frac{4K^2\eta_l^2L^2}{N}\right)\sum_{i=1}^N\mathbb{E}[\|\boldsymbol{\theta}_{t,0}^i-\boldsymbol{\theta}_t\|^2]$$

$$+(120K^4\eta_l^4L^2+4K^2\eta_l^2)\mathbb{E}[\|\nabla_{b_t}f(\boldsymbol{\theta}_t)\|^2]$$

$$\leq K\eta_l^2\sigma^2+10K^3\eta_l^4L^2\sigma^2+60K^4\eta_l^4L^2\sigma_g^2+(240K^4\eta^2\eta_l^4L^4+8K^2\eta^2\eta_l^2L^2)\frac{1}{N}\sum_{i=1}^N\mathbb{E}[\|\boldsymbol{\Delta}_{t-1}^i\|^2]$$

$$+(240K^4\eta^2\eta_l^4L^4+8K^2\eta^2\eta_l^2L^2)\mathbb{E}[\|\boldsymbol{\Delta}_{t-1}\|^2]+(120K^4\eta_l^4L^2+4K^2\eta_l^2)\mathbb{E}[\|\nabla_{b_t}f(\boldsymbol{\theta}_t)\|^2]. \tag{39}$$

First we previously assume that $\eta_l\leq\frac{1}{8KL}$, and also (1) for simplicity, if we have a sequence $x_t\leq\alpha x_{t-1}+\alpha y_{t-1}+\beta z_t+C$, then we have

$$x_t\leq\alpha x_{t-1}+\alpha y_{t-1}+\beta z_t+C$$
$$\leq\alpha(\alpha x_{t-2}+\alpha y_{t-2}+\beta z_{t-1}+C)+\alpha y_{t-1}+\beta z_t+C$$
$$\cdots$$
$$\leq\alpha^t x_0+\sum_{i=1}^t\alpha^i y_{i-1}+\sum_{i=1}^t\alpha^{i-1}\beta z_i+C\sum_{i=1}^t\alpha^{i-1}.$$

(2) for simplicity, if we have a sequence $x_t \leq \alpha x_{t-1} + \alpha y_{t-1} + \beta z_t + C$, then we have

$$x_t \leq \alpha x_{t-1} + \alpha y_{t-1} + \beta z_t + C$$

$$\sum_{t=0}^{T-1} x_t \leq \alpha \sum_{t=0}^{T-1} x_{t-1} + \alpha \sum_{t=0}^{T-1} y_{t-1} + \beta \sum_{t=0}^{T-1} z_t + C * T$$

$$\Rightarrow$$

$$\sum_{t=0}^{T-1} x_t \leq \alpha \sum_{t=0}^{T-1} x_t + \alpha \sum_{t=0}^{T-1} y_{t-1} + \beta \sum_{t=0}^{T-1} z_t + C * T$$

$$(1-\alpha) \sum_{t=0}^{T-1} x_t \leq \alpha \sum_{t=0}^{T-1} y_{t-1} + \beta \sum_{t=0}^{T-1} z_t + C * T$$

$$\sum_{t=0}^{T-1} x_t \leq \alpha(1-\alpha)^{-1} \sum_{t=0}^{T-1} y_{t-1} + \beta(1-\alpha)^{-1} \sum_{t=0}^{T-1} z_t + C(1-\alpha)^{-1} * T,$$

we want that $\frac{1}{2} \leq 1 - \alpha < 1$, which means $0 < \alpha \leq \frac{1}{2}$ therefore, we have $1 < (1-\alpha)^{-1} \leq 2$. Moreover, since $\alpha = 240K^4\eta^2\eta_l^4 L^4 + 8K^2\eta^2\eta_l^2 L^2 \leq \frac{1}{2}$, we have $240K^4\eta_l^4 L^2 + 8K^2\eta_l^2 \leq \frac{1}{2\eta^2 L^2}$

$$\sum_{t=0}^{T-1} x_t \leq \sum_{t=0}^{T-1} y_t + 2\beta \sum_{t=0}^{T-1} z_t + 2C * T$$

$$\Rightarrow \tag{40}$$

$$\sum_{t=0}^{T-1} \left[ \frac{1}{N} \sum_{i=1}^{N} \mathbb{E}[\|\boldsymbol{\Delta}_t^i\|^2] \right] \leq \sum_{t=0}^{T-1} \mathbb{E}[\|\boldsymbol{\Delta}_t\|^2] + (240K^4\eta_l^4 L^2 + 8K^2\eta_l^2) \sum_{t=0}^{T-1} \mathbb{E}[\|\nabla_{b_t} f(\boldsymbol{\theta}_t)\|^2]$$

$$+ 2TK\eta_l^2\sigma^2 + 20TK^3\eta_l^4 L^2\sigma^2 + 120TK^4\eta_l^4 L^2\sigma_g^2$$

$$\leq \sum_{t=0}^{T-1} \mathbb{E}[\|\boldsymbol{\Delta}_t\|^2] + \frac{1}{2\eta^2 L^2} \sum_{t=0}^{T-1} \mathbb{E}[\|\nabla_{b_t} f(\boldsymbol{\theta}_t)\|^2]$$

$$+ 2TK\eta_l^2\sigma^2 + 20TK^3\eta_l^4 L^2\sigma^2 + 120TK^4\eta_l^4 L^2\sigma_g^2. \tag{41}$$

$\square$

**Lemma C.3.** *For $\boldsymbol{\theta} = [\boldsymbol{\theta}^1, \boldsymbol{\theta}^2, \ldots, \boldsymbol{\theta}^B]$, i.e., there is a block partition $b = 1, 2, \ldots, B$ partitioned $\boldsymbol{\theta}$ into $B$ blocks, then we have $\|\boldsymbol{\theta}\|^2 = \sum_{b=1}^{B} \|\boldsymbol{\theta}^b\|^2$.*

*Proof.*

$$\|\boldsymbol{\theta}^1\|^2 + \|\boldsymbol{\theta}^2\|^2 + \cdots + \|\boldsymbol{\theta}^B\|^2$$

$$= \left( \sum_{i=1}^{d_1} (x^{1,i})^2 \right) + \left( \sum_{i=1}^{d_2} (x^{2,i})^2 \right) + \cdots + \left( \sum_{i=1}^{d_B} (x^{B,i})^2 \right)$$

$$= \sum_{i=1}^{d} (x^i)^2 = \|\boldsymbol{\theta}\|^2. \tag{42}$$

$\square$

**Lemma C.4.** *For $\boldsymbol{\theta} = [\boldsymbol{\theta}^1, \boldsymbol{\theta}^2, \ldots, \boldsymbol{\theta}^B]$ and $\boldsymbol{y} = [\boldsymbol{y}^1, \boldsymbol{y}^2, \ldots, \boldsymbol{y}^B]$, i.e., there is a block partition $b = 1, 2, \ldots, B$ partitioned $\boldsymbol{\theta}$ and $\boldsymbol{y}$ into $B$ blocks, then we have $\langle \boldsymbol{\theta}, \boldsymbol{y} \rangle = \sum_{b=1}^{B} \langle \boldsymbol{\theta}^b, \boldsymbol{y}^b \rangle$.*

*Proof.*

$$\langle \boldsymbol{\theta}^1, \boldsymbol{y}^1 \rangle + \langle \boldsymbol{\theta}^2, \boldsymbol{y}^2 \rangle + \cdots + \langle \boldsymbol{\theta}^B, \boldsymbol{y}^B \rangle$$

$$= \sum_{i=1}^{d_1} x^{1,i} y^{1,i} + \sum_{i=1}^{d_2} x^{2,i} y^{2,i} + \cdots + \sum_{i=1}^{d_B} x^{B,i} y^{B,i}$$

$$= \sum_{i=1}^{d} x^i y^i = \langle \boldsymbol{\theta}, \boldsymbol{y} \rangle. \tag{43}$$

$\square$

