# OpenReview forum: "ParaBlock: Communication-Computation Parallel Block Coordinate Federated Learning for Large Language Models"
_TMLR — Accepted by TMLR_

### Review · Reviewer_gZ8W · 2025-11-14

**Summary Of Contributions:**

This paper proposes to parallelize training and parameter exchange in federated learning to reduce latency caused by exchanging large parameter vectors over a network. The parallelization is enabled by a block gradient descent scheme, where each training run between parameter exchange only trains on a subset of the parameters.

**Additional Comments:**

I recommend that the authors thoroughly revise the paper and resubmit.

**Audience:**

No

**Audience Explanation:**

Federated learning with large parameter models has become an increasingly relevant field since the advent of large transformer models.

**Broader Impact Concerns:**

None at this point in time.

**Claims And Evidence:**

No

**Claims Explanation:**

The main theoretical result, Theorem 5.3, differs from the result that is proven in the Appendix contained in the supplementary material. The result in the Appendix does not contain the constant $\mathcal{F}$ but instead depends on the difference between the parameter initialization and the expected value of the final parameters. Furthermore, the constants in the Appendix and in Theorem 5.3 do not match.

The explanation of the Algorithm is also flawed, in particular, Figure 1.

**Requested Changes:**

- Fix Theorem 5.3 or the respective proof.
- Figure 1 displays that the clients receive parameter updates downstream before having sent any parameter updates. That does not make sense in this setup.
- The recursive step in Equation (3) is unclear. It is also unclear to me what this discussion is trying to show. If the goal is only to show that the local parameters in block $b_{t-1}$ are equal to the global parameters after two iterations, the discussion is unnecessarily convoluted.
- The presentation needs to be improved. Examples from the first page only:
  * "... the proposed ParaBlock achieves ..." What part of ParaBlock are you referring to?
  * "Federated learning is known as a promising privacy-preserving solution" ... solution to what?
  * Gemma 2 was not published by an individual with last name "Team" and their collaborators, but the Qwen Team, without further contributors.

---

> ### Author Response · Authors · 2026-01-12
> **Thank you for the review**
>
> **Q1.** Regarding the constant in Theorem 5.3
>
> Thank you for your detailed review and valuable comments on our theoretical results. We apologize for the confusion caused by the omission of intermediate steps regarding term combination and variable substitution in the Appendix.
>
> * Regarding the definition of $F$. In summarizing our results to obtain Equation (27), we utilized the definition provided in Theorem 5.3. Here, $F$ represents the difference between the loss at parameter initialization and the minimum expected loss, defined as $F=f(\theta_0)-f_*$, where $f_*= min_{\theta} f(\theta)$. Since $E[f(\theta_T)] \ge E[f_*]$, we have the relationship: $f(\theta_0)-E[f(\theta_T)] \leq f(\theta_0)-E[f_*] =F$. By applying this substitution, Equation (27) yields the final bound presented in Theorem 5.3.
> * Regarding the constants. We apologize for the lack of clarity here. The constant in Equation (27) is indeed consistent with that in Theorem 5.3. The apparent discrepancy arose because we performed algebraic simplifications and grouped similar constant terms in the final statement of Theorem 5.3.
>
> We have revised the Appendix and to include these missing steps and clarifications.
>
> **Q2.** Regarding Figure 1.
>
> We wish to clarify that Figure 1 is intended as a schematic representation of the protocol's operations, rather than a complete process of the entire timeline.
>
> The depiction of the downlink in Block 1 implies the existence of preceding steps (such as server-side initialization or updates from previous rounds). To make this clear and indicate that this block is part of a continuous sequence, we have updated Figure 1 to include ellipses (...) at both the beginning and the end of the timeline.
>
> **Q3.** Regarding the recursive step in Equation (3).
>
> We apologize if the derivation appeared convoluted. We clarify that Equation (3) serves two primary purposes. First, it demonstrates that the local blocks align with the global parameters after correction (as noted in your review). Second, it derives the specific expression of the local parameters, which serves as a fundamental building block for our convergence analysis.
>
> Specifically, this equation establishes the exact mathematical form required to bound the variance term in our convergence proof. Without this explicit recursive formulation, it would be difficult to rigorously justify how the one-round delayed correction guarantees the algorithm's convergence speed.
>
> **Q4.** Regarding the presentation.
>
> We sincerely thank the reviewer for the careful reading and for pointing out these presentation issues. We apologize for the imprecise phrasing and the citation errors.
>
> Regarding the citations for Gemma 2 and Qwen, we wish to clarify that we directly utilized the official BibTeX entries provided in their original releases, which formatted the authors as author={Team, Gemma ...} and author={Qwen Team} respectively. This caused standard BibTeX styles to misinterpret "Team" as a surname (e.g., outputting "Team, Q.").
>
> Gemma 2 paper: https://arxiv.org/abs/2408.00118
>
> Gemma 2 bib:
> ```
> @article{team2024gemma,
>   title={Gemma 2: Improving open language models at a practical size},
>   author={Team, Gemma and Riviere, Morgane and Pathak, Shreya and Sessa, Pier Giuseppe and Hardin, Cassidy and Bhupatiraju, Surya and Hussenot, L{\'e}onard and Mesnard, Thomas and Shahriari, Bobak and Ram{\'e}, Alexandre and others},
>   journal={arXiv preprint arXiv:2408.00118},
>   year={2024}
> }
> ```
> Qwen 2.5 blog: https://qwenlm.github.io/blog/qwen2.5/
>
> Qwen 2.5 official bib:
> ```
> @misc{qwen2.5,
>     title = {Qwen2.5: A Party of Foundation Models},
>     url = {https://qwenlm.github.io/blog/qwen2.5/},
>     author = {Qwen Team},
>     month = {September},
>     year = {2024}
> }
> ```
> We have thoroughly proofread and revised the manuscript to address other concerns accordingly.

---

### Review · Reviewer_WqtU · 2025-11-17

**Summary Of Contributions:**

The authors present ParaBlock, a federated supervised learning method particularly useful for very large models such as LLMs. For such large models, communication of the parameters can cause a significant slowdown for existing federated learning methods.   ParaBlock uses concurrent communication and computation threads to speed up the overall learning process so that communication and computation overlap better. Appropriate algorithms are designed for updating the current block as well as "catching up" the previously updated block at each round. Theoretical asymptotic convergence results in terms of the number of global rounds are obtained. Largely superior performance results are obtained in terms of speed and accuracy in comparison with competing methods on some LLM benchmarks. Ablation experiments are also presented.

The paper is certainly relevant to today's cutting edge AI concerns, although the utility of federated learning for LLMs could be better explained (I will elaborate in a later section). Industrial-strength LLM models are now enormous and any speedup algorithms should be of widespread interest.

The method makes a lot of intuitive sense and the experimental results are reasonably thorough and convincing, although the details of the 'full fine tuning' benchmark and the interpretation of the comparison in speed and performance between 'full fine tuning' and ParaBlock needs more explanation.

The asymptotic theoretical convergence results are of debatable significance, in my opinion, since it's not clear to me whether there are enough global rounds T to make asymptotics relevant.

**Additional Comments:**

Some typos and suggested rewrites:

P2 by cyclically elects -> by cyclically electing
P2 schemes enhances -> schemes enhance
P 2 have demonstrated to enhance -> have been shown to enhance
P3 descent, it. -> descent. It (new sentence needed)
P3 substituting the SGD optimizer to BCD -> substituting BCD for the SGD optimizer
P4 are proceed -> proceed
P5 Alg 2 step 7 client I sends, waits (not send, wait)
P5 This demonstrated -> This demonstrates
P7 and an lightweight -> and a lightweight
P3 Approximates to optimize -> approximately optimizes
P10 various block partition-> various block partitions

**Audience:**

Yes

**Audience Explanation:**

There should be plenty of interest in this paper. Very large scale LLMs are of widespread interest and speedup is of their training is of great interest.

**Broader Impact Concerns:**

I'm not very worried here but I am slightly worried. I'm neither an expert on LLMs nor federated learning, but it's not intuitively obvious to me that federated learning can preserve privacy in an LLM in the desired way. There's clearly been a fair amount of work on federated learning for LLMs, judging from the paper references, so I guess I will trust in the research community here.  Still, it's not obvious to me that specific facts from text within one silo of the federation, where privacy is a concern, won't end up being regurgitated by the final LLM in a response to a user query. I don't see how you could guarantee that.

**Claims And Evidence:**

Yes

**Claims Explanation:**

It's honestly borderline for me between 'yes' and 'no'. Some clarifications are needed before publication, as I will explain in the next section, but on the balance, I would say yes.

**Requested Changes:**

The 'full fine-tuning' benchmark and its performance and speed as compared to ParaBlock needs to be better explained.

Is 'full fine-tuning' no federation at all, i.e., the entire model being trained with full, granular access to all the training data? If there's no federation at all, why would it ever lose to ParaBlock, as it does in terms of both accuracy and speed on Llama3.2-3B in Table 1?

Somewhat relatedly, at the start of page 8, the paper says ParaBlock achieves a lower score in the 'main middle columns' of Table 1. However, it's actually the first 2 columns on the left, for Llama3-8B,  not the middle ones, where ParaBlock gets a lower score. ParaBlock gets a higher score in the middle column, for Llama3.2-3B. This seems to be a typo that needs fixing, unless I've misunderstood something.

Page 7 says that 32 and 28 global rounds T of fine-tuning were used. Theorem 5.3 and Corollary 5.4 have bounds in terms of T. Is T ~30 large enough to appeal to the asymptotic convergence behavior? The constants up front in equation 5 are pretty big (64, 40, etc). Sometimes, a bound that appears useful at first glance may not be that useful when you plug in realistic numbers. I would like more reassurance that the bound is useful given real-life values for the variables in the bound.

---

> ### Author Response · Authors · 2026-01-12
> **Thank you for the review**
>
> **Q1.** Comparison and discussion about full fine-tuning.
>
> We apologize for the confusion.
> * We confirm that the "full fine-tuning" baseline refers to **Federated Full Fine-tuning**, not centralized training. In this setup, clients strictly keep data local and perform full parameter updates. Therefore, it is subject to the same challenges as other federated methods.
> * Regarding the speed. While Full Fine-Tuning (Full FT) requires fewer global rounds to converge (e.g., ~3 rounds vs. ~30 for PEFT methods), its communication overhead per round is prohibitive as it involves transmitting the entire model. Consequently, the speed advantage of ParaBlock stems from alleviating this communication bottleneck.
> In contrast to Full FT, ParaBlock transmits only a small subset of parameters and utilizes two-thread parallelism to overlap communication with computation. Therefore, the total runtime comparison depends on the following trade-off:
>     * If communication latency is significant or the total computation time for Full FT is not significantly shorter than ParaBlock, the heavy communication overhead of Full FT outweighs its advantage in round efficiency, making ParaBlock faster (as shown in some cases in Table 1).
>     * Conversely, Full FT would be faster in scenarios with substantial computational savings and sufficient bandwidth (where the communication overhead is manageable).
>
> * Regarding the better performance of ParaBlock than full fine-tuning. This is indeed an interesting phenomenon that has also been observed in other PEFT methods. For example, LoRA-based approaches such as LoRA-GA[1]and LoRA-Pro[2] sometimes outperform full-model fine-tuning, and sometimes for the layer-wise method, such as LISA[3], on certain tasks. We believe this can be attributed to implicit regularization effects and better generalization due to fewer trainable parameters in each round. In our case, ParaBlock’s selective block updates may help prevent overfitting and allow the model to focus learning on the submodules during each round, leading to improved downstream performance in some settings.
>
> [1] Wang, S., Yu, L., & Li, J. (2024). Lora-ga: Low-rank adaptation with gradient approximation. Advances in Neural Information Processing Systems.
>
> [2] Wang, Z., Liang, J., He, R., Wang, Z., & Tan, T. (2024). Lora-pro: Are low-rank adapters properly optimized?. arXiv preprint arXiv:2407.18242.
>
> [3] Pan, R., Liu, X., Diao, S., Pi, R., Zhang, J., Han, C., & Zhang, T. (2024). Lisa: Layerwise importance sampling for memory-efficient large language model fine-tuning. Advances in Neural Information Processing Systems.
>
> We have revised the paper to add discussions accordingly.
>
> **Q2.** Regarding the description of Table 1 and Page 8.
>
> We apologize for the misleading description. The original 'main middle columns' of Table 1 refer to the results trained on Alpaca-GPT4 datasets. What we want to emphasize is that "ParaBlock achieves a **lower score than Full FT** but consistently outperforms most LoRA-based and BCD-based
> PEFT methods across two models (8B and 3B models)." We have revised the description to make the content clear.
>
> **Q3.** Regarding the global round number T.
>
> We thank the reviewer for this insightful comment. We would like to clarify the significance of the bound from two perspectives: the general theoretical framework and the specific experimental context.
>
> First, our theoretical contribution (Theorem 5.3) is designed to establish the convergence guarantee for the general ParaBlock framework, applicable to a wide range of non-convex optimization problems (including training from scratch and fine-tuning).
> * The primary value of the derived bound is to verify the convergence behavior and explicitly characterize how the convergence rate scales with key factors like the number of rounds $T$, clients $N$, and local steps $K$.
> * The large constants are derived from worst-case scenarios, thus serve to ensure theoretical rigor across general cases, rather than to provide precise numerical predictions for specific experiments.
> * Moreover, in training-from-scratch scenarios, $T$ is typically much larger than 30, often reaching hundreds or even thousands. Even in the fine-tuning context ($T \approx 30$), the primary goal of our analysis is to demonstrate that ParaBlock achieves the same convergence rate as existing Federated BCD methods (e.g., FedBCD and FedBCGD) under general non-convex settings. (The convergence bounds for these baseline methods also involve large constants similar to ours.) Our experiments empirically verify this alignment. As shown in Figure 2c, ParaBlock and FedBCD exhibit very similar convergence behaviors; while ParaBlock shows marginally slower convergence in the initial stages, this gap diminishes significantly in later rounds. These empirical results support our theoretical findings.

---

> > ### Author Response · Authors · 2026-01-12
> >
> > **Q4.** Broader Impact Concerns.
> >
> > We acknowledge the reviewer's thoughtful consideration of the broader privacy implications. While FL ensures data ownership protection (raw data stays local), it does not automatically prevent the model from learning and outputting sensitive patterns. This privacy protection is a known challenge in generative language models.
> >
> > Our work primarily addresses the efficiency and communication bottlenecks of fine-tuning LLMs on edge devices. However, we recognize that for real-world deployment where privacy preserving is critical, our framework should be deployed alongside techniques like data anonymization or differential privacy to minimize the risk of data privacy leakage. We have added the discussion in the manuscript.
> >
> > Thanks for mentioning the typo and minor presentation issues. We have fixed that in our revision.

---

### Review · Reviewer_TsJE · 2026-01-04

**Summary Of Contributions:**

This paper proposes ParaBlock, a parallel client-side execution framework for federated block coordinate descent in LLM fine-tuning. By overlapping local computation and communication via dual threads and introducing a staleness-aware correction mechanism, the method reduces idle waiting without altering the core optimization structure. The authors provide convergence guarantees comparable to standard federated BCD and demonstrate that ParaBlock achieves lower wall-clock training time while maintaining competitive performance across multiple LLM fine-tuning tasks under varying communication conditions.

**Audience:**

Yes

**Audience Explanation:**

The paper would interest parts of the TMLR audience working on federated learning, LLM fine-tuning, and communication-efficient optimization, as it addresses a practical and timely systems bottleneck.

**Claims And Evidence:**

Yes

**Claims Explanation:**

The core claims are generally supported by clear experimental results showing reduced wall-clock time and comparable task performance to federated BCD baselines. The algorithm and correction mechanism are described clearly, and the efficiency gains are consistently demonstrated under different bandwidth settings.

**Requested Changes:**

- All experiments use only 10 clients, which is far from realistic federated learning settings with hundreds or thousands of clients. Consequently, the claimed benefits of ParaBlock in large-scale FL are not fully supported, and the effect of one-round staleness at scale remains unclear. Moreover, the evaluation assumes fixed, homogeneous bandwidths (50/100/150 Mbps) and does not consider network heterogeneity or stragglers. Additional clarification would strengthen this part.
- The convergence analysis assumes local SGD, while all experiments use AdamW, resulting in a theory–practice mismatch that should be clarified or addressed.
- The baselines mainly include LoRA-based methods and FedCyBGD, but recent asynchronous federated learning approaches also address communication waiting. Including such baselines would provide a more comprehensive evaluation.
- The authors should consider improving the reproducibility of the work by providing additional experimental details, such as bandwidth simulation, client configuration, and random seeds.

---

> ### Author Response · Authors · 2026-01-12
> **Thank you for the review**
>
> **Q1. a)** Regarding the number of clients and the effect of one-round staleness.
>
> We appreciate the reviewer's concern regarding the experimental scale.
> * First, we respectfully clarify that our primary focus is cross-silo federated learning (e.g., collaboration between financial institutions or hospitals). In such settings, the number of participants is naturally limited (typically 5–20), but each client possesses substantial computing and data resources. Furthermore, simulating federated LLMs fine-tuning is resource-intensive. Our initial setup with 10 clients was chosen to balance realistic cross-silo simulation with the significant GPU memory constraints.
>
> * Note that the effect of one-round staleness is not directly related to the number of participating clients $N$.
>     * The staleness in ParaBlock is strictly due to the latest training happening on previous parameters, i.e., using parameters from $t-1$ at round $t$.
>     * This "staleness" remains constant (exactly one round) regardless of whether the aggregation involves 10 or 1000 clients. The error correction mechanism (Eq. 3) functions identically irrespective of the system scale, and the "one-round staleness" does not compound or degrade as the number of clients increases.
>
> To further address the reviewer's concern about scalability and confirm that our findings hold beyond $N=10$, we have conducted additional experiments scaling the number of clients to $N=100$. (Note that in our initial Table 7, we have already provided one basic experiment of cross-silo settings with $N=50$ clients in total.)
>
> The following results demonstrate that ParaBlock maintains its performance advantage with $N=100$ clients in cross-silo settings.
>
> | Method | GSM8K | Runtime(m) |
> | -------- | -------- | -------- |
> | FedIT     | 28.28 | 25.6 |
> | FedBCD | 28.35 | 21.3 |
> | ParaBlock | 29.57 | 15.7 |
>
> b) Regarding homogeneous bandwidth.
> We adjust our algorithm to heterogeneous communication bandwidth. For example, we conducted an experiment in which one client was significantly slower than the others. The extremely slow client incurs a three-round staleness, while other clients maintain the original one-round of staleness. The following results illustrate this scenario. Note that in this heterogeneous bandwidth setting, slow clients with significant latency may degrade overall performance. We believe it requires a non-trivial design for better performance. We have added discussions in our revision.
>
> | | Math Instruct (GSM8K) | Runtime |
> | -------- | -------- | -------- |
> | ParaBlock (adapted)     | 52.46     | 20.16     |
> | ParaBlock (orignial) | 55.88 | 38.72
>
>
> **Q2.** The convergence analysis with AdamW.
> * The goal of our convergence analysis is to demonstrate that the design of ParaBlock--with one-round staleness--does not compromise the convergence rate. To this end, we compare our results with existing analyses of FedBCD, which are based on local SGD updates (Liu et al., 2019; Wu et al., 2021). Thus, we also adopt local SGD in our theoretical analysis for both convenience and fair comparison. We will clarify this point in the revised version of the paper.
> * Our theoretical analysis is definitely not limited to the local SGD setting. We originally included the discussion and adaptation to a local adaptive optimizer, such as AdamW, in our Appendix B.3. We have also pointed this out in the main paper.
>
> **Q3.** Comparison with asynchronous FL baselines.
>
> * First, we would like to clarify that ParaBlock is significantly different from asynchronous FL, although both share the motivation of addressing communication waiting and efficiency.
>     * The efficiency in asynchronous FL is letting clients update to the server at their own pace, so it may happen that one client is conducting communication, but another client is computing the gradient.
>     * While Parablock is setting up a two-thread parallel scheme for every single client.
> * Thus, it may not be fair to directly compare asynchronous FL approaches with ParaBlock, as they are designed for full-parameter training. Nevertheless, we compare one representative asynchronous FL baseline, FedBuff, with ParaBlock for fine-tuning the LLaMA-3 8B model on mathematical reasoning tasks. In a setting where 2 out of 10 clients experience extreme delays, and the FedBuff algorithm is applied, the GSM8K score drops to 52.24, which is significantly lower than ParaBlock’s score of 55.88. We believe this degradation is due to the impact of slow clients in FedBuff. While asynchronous training remains a promising direction, it would require a non-trivial redesign to be effective.
>
> We have added discussions in our revision.

---

> > ### Author Response · Authors · 2026-01-12
> >
> > **Q4.**  Additional experimental details.
> > We thank the reviewer for emphasizing the importance of reproducibility. We have included a hyper-parameter details section in Appendix A.1. Moreover, we have revised the main paper to elaborate on the experimental details. To sum up:
> > * Due to deployment constraints, we simulated the communication bandwidth with three network bandwidth conditions (50 MB/s, 100 MB/s, and 150 MB/s), these were chosen based on frequent settings in FL and decentralized learning. Then we calculate the communication time based on the size of the model parameters to be transmitted.
> >
> > * We have specified the hardware and software environment details. All experiments were conducted on NVIDIA A100 GPUs. Given the fixed hardware setting (e.g., a single A100 node for a client), the training time for local fine-tuning was measured based on the actual runtime of the A100 GPU. Additionally, by adjusting the batch size and gradient accumulation steps, we varied the computation time on clients.
> > * We fixed the random seeds for all libraries to 42. To fully support reproducibility, our code will be released later.

---

### Decision · Action_Editor_NSua · 2026-02-20

**Recommendation:** Accept with minor revision

**Additional Comments:**

Please further check the reviewers' suggestions about the presentation issues and fix that for the final publication.

**Audience:**

Yes

**Audience Explanation:**

This is a relevant topic about federated learning and LLMs for the machine learning community.

**Claims And Evidence:**

Yes

**Claims Explanation:**

The experimental results look solid, and the concern about the math proof has been fixed. Although some presentation issues require further refinement for the final publication, the statements made by the authors in this paper are generally convincing.

**Resubmission Of Major Revision:**

The authors may consider submitting a major revision at a later time.